# A second DNA binding site on RFC facilitates clamp loading at gapped or nicked DNA

Xingchen Liu[†], Christl Gaubitz*[†], Joshua Pajak, Brian A Kelch*

Department of Biochemistry and Molecular Biotechnology, University of Massachusetts Chan Medical School, Worcester, United States

**Abstract** Clamp loaders place circular sliding clamp proteins onto DNA so that clamp-binding partner proteins can synthesize, scan, and repair the genome. DNA with nicks or small single-stranded gaps are common clamp-loading targets in DNA repair, yet these substrates would be sterically blocked given the known mechanism for binding of primer-template DNA. Here, we report the discovery of a second DNA binding site in the yeast clamp loader replication factor C (RFC) that aids in binding to nicked or gapped DNA. This DNA binding site is on the external surface and is only accessible in the open conformation of RFC. Initial DNA binding at this site thus provides access to the primary DNA binding site in the central chamber. Furthermore, we identify that this site can partially unwind DNA to create an extended single-stranded gap for DNA binding in RFC's central chamber and subsequent ATPase activation. Finally, we show that deletion of the BRCT domain, a major component of the external DNA binding site, results in defective yeast growth in the presence of DNA damage where nicked or gapped DNA intermediates occur. We propose that RFC's external DNA binding site acts to enhance DNA binding and clamp loading, particularly at DNA architectures typically found in DNA repair.

*For correspondence:
cgaubitz@sund.ku.dk (CG);
brian.kelch@umassmed.edu
(BAK)

[†]These authors contributed
equally to this work

**Competing interest:** The authors declare that no competing interests exist.

## Editor's evaluation

This study leverages the fortuitous finding that the eukaryotic RFC clamp loader binds to two DNA molecules at distinct sites. The first at the previously described internal binding site and the second, novel, site at a site on RFC1 external to the loader and above (viewed from a certain perspective) the PCNA ring. The authors extend and validate their chance finding using biochemical, genetic, and structural studies. The cryo-EM structures of RFC bound to nicked DNA that emerge from this study write a new chapter in the biochemistry of clamp loading.

## Introduction

Sliding clamps are ring-shaped proteins that surround DNA to tether DNA polymerases and other factors onto DNA. The sliding clamp of eukaryotes, proliferating cell nuclear antigen (PCNA), can bind to scores of different partners and facilitates numerous pathways such as DNA replication, DNA repair, cell cycle control, and chromatin structure (*Choe and Moldovan, 2017*; *Moldovan et al., 2007*). Thus, PCNA is a highly adaptable protein because it can interact with a diverse array of protein partners and DNA substrates.

Because sliding clamps are closed in solution, they must be loaded onto DNA by clamp loader ATPase complexes. Clamp loaders, like sliding clamps, are conserved throughout all life and are necessary for proper replication and regulation of the genome. Clamp loaders are members of the *ATPases Associated with various cellular Activities* (AAA+) family of ATPases and are related to many other

important macromolecular machines that regulate DNA replication, such as DNA replication initiators, helicases, and helicase loaders (*Erzberger and Berger, 2006*). Unlike typical AAA+ machines that function as hexamers, clamp loaders are all pentameric ATPases. Each subunit is termed A through E and consists of a AAA+ module connected to a C-terminal collar domain that oligomerizes the complex.

Because clamp loaders function in numerous pathways, they can utilize diverse nucleic acid substrates. The classical DNA substrate that clamp loaders use for DNA replication is primer-template DNA (p/t-DNA), a duplex region with a 5′ single-strand DNA (ssDNA) overhang and a recessed 3′ end (*Ason et al., 2003*; *Tsurimoto and Stillman, 1991*). By loading a clamp onto p/t-DNA, clamp loaders position the sliding clamp to be subsequently utilized by a DNA polymerase to extend the primer strand 3′ end (*Naktinis et al., 1996*). Structures of p/t-DNA-bound clamp loaders from archaea, bacteria, viruses, and eukaryotes have revealed that the mechanism of p/t-DNA recognition is broadly conserved (*Gaubitz et al., 2021*; *Kelch et al., 2011*; *Miyata et al., 2005*; *Simonetta et al., 2009*). The duplex region of p/t-DNA is bound within the central chamber of the clamp loader with the 3′ recessed end capped by the collar region. We recently reported that the eukaryotic clamp loaders have a 'separation pin' that can melt the 3′ base pair of the primer strand (*Gaubitz et al., 2021*). The ssDNA overhang of the template strand exits the central chamber through a gap in between the A and A′ domains that is termed the 'A-gate.' The A-gate plays a pivotal role in clamp loader function because it allows entry and egress of DNA from the clamp loader central chamber. We recently showed that the opening of the A-gate requires a massive conformational change from a closed, autoinhibited state into an open, active conformation (*Gaubitz et al., 2020*; *Gaubitz et al., 2021*). Moreover, the A-gate provides specificity for a ssDNA region because the gap at the top of the central chamber is too kinked and narrow for dsDNA (*Kelch et al., 2011*; *Gaubitz et al., 2021*).

However, clamp loaders also utilize many DNA architectures other than p/t-DNA, including small ssDNA gaps and even nicked DNA with no ssDNA (*Matsumoto, 2001*; *Uhlmann et al., 1997*). These substrates are more commonly seen in DNA repair pathways such as long patch base excision repair (lp-BER) (*Matsumoto, 2001*). Our recent structures of the yeast clamp loader replication factor C (RFC) bound to p/t-DNA revealed that a 'separation pin' in RFC melts the 3′ end of primer. Although the function of this activity remains obscure, we hypothesize that the separation may be involved in binding different DNA architectures (*Gaubitz et al., 2021*). Moreover, nonclassical clamp loaders called RFC-like complexes (RLCs), such as Rad24-RLC and Elg1-RLC, utilize DNA structures with recessed 5′ ends or fully dsDNA, respectively (*Kubota et al., 2015*; *Majka and Burgers, 2004*; *Majka et al., 2006*; *Ellison and Stillman, 2003*). Recent structures of Rad24-RLC show that the 5′ recessed DNA is surprisingly accommodated on a second DNA binding site on the 'shoulder' of Rad24 (*Castaneda et al., 2021*; *Zheng et al., 2021*). However, the structures of RFC and Rad24-RLC do not indicate how gapped or nicked DNA can bind to RFC. Moreover, it is not known how clamp loaders evolved different DNA binding specificity.

Here, we describe a series of structures that reveal that RFC has a second binding site for DNA on the external surface of RFC. This external DNA binding site is composite, consisting of three domains of Rfc1: the (BRCA1 C-Terminus) BRCT, AAA+, and collar domains. Furthermore, we observe that this site is used to bind gapped or nicked DNA. Cryo-EM structures of RFC:PCNA bound to these DNA architectures reveal that the external DNA binding site is positioned to interact with the 5′ duplex region of gapped or nicked DNA. To bind nicked DNA, the clamp loader must melt DNA at both the interior and exterior DNA binding sites, indicating that RFC has two separation pins. Finally, we show that yeast specifically lacking the BRCT region of the external binding site exhibit a phenotype consistent with disrupted BER. Our results reveal a region of RFC used for binding DNA architectures that commonly appear during DNA repair, and we link the evolution of altered DNA binding in Rad24-RLC to the classical clamp loader RFC.

## Results

### Structure of RFC:PCNA:p/tDNA reveals a second DNA binding site

We previously described a series of structures of yeast RFC bound to PCNA, ATPγS, and p/t- DNA (*Gaubitz et al., 2021*). Here, we describe a class at 3.4 Å overall resolution from one of these datasets that contains a second DNA bound at a previously undescribed site (*Figure 1A and B*,

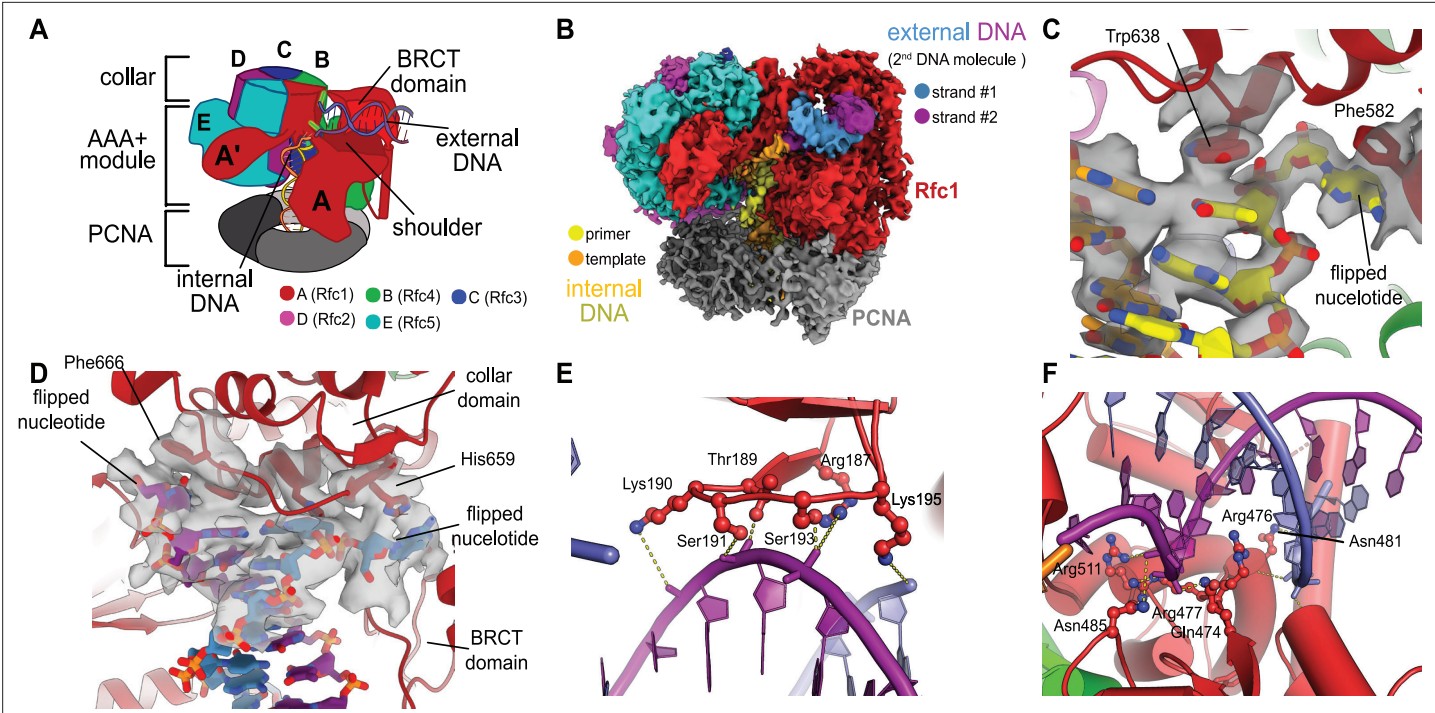

**Figure 1.** Structure of replication factor C:proliferating cell nuclear antigen (RFC:PCNA) bound to two p/t-DNA molecules. (**A**) Schematic of the complex of RFC:PCNA bound to two p/t-DNAs. Melted base pairs are shown as glowing green sticks. (**B**) Cryo-EM reconstruction of the complex of RFC:PCNA bound to two p/t-DNAs. The strands of the external DNA are shown in slate and purple coloring. (**C**) The 3' nucleotide of the primer strand is melted at the internal separation pin. (**D**) The external DNA binding site also melts DNA. The two melted bases stack against Phe666 and His659. (**E**) The BRCT domain grips duplex DNA at the external DNA binding site. (**F**) The 'shoulder' region of the AAA+ module grips duplex DNA, with Gln474 and Arg476 inserting into the minor groove, presumably setting the register of DNA.

The online version of this article includes the following source data and figure supplement(s) for figure 1:

**Figure supplement 1.** Cryo-EM processing of replication factor C:proliferating cell nuclear antigen (RFC:PCNA) in the presence of p/t DNA.

**Figure supplement 2.** Structural similarity between structures with one or two p/t-DNAs bound.

**Figure supplement 3.** Conservation of key residues for binding DNA.

**Figure supplement 4—source data 1.** 2-Aminopurine fluorescence.

*Figure 1—figure supplement 1*). The primary p/t-DNA binding site is filled normally, with the nucleotide at the 3' end of the primer flipped by the previously described internal separation pin (*Gaubitz et al., 2021*; *Figure 1C*; *Figure 1—figure supplement 2*). A second p/t-DNA molecule fortuitously binds to RFC on the outside of the complex (*Figure 1A and B*). The second DNA binding site is just outside the exit channel for the single-stranded template overhang at the A-gate. The second DNA binding site is comprised entirely by the A subunit (Rfc1) at the 'shoulder region,' which is at the top of AAA+ module and next to the collar domain. We observe nine base pairs of duplex DNA, and a melted base pair in between the collar and BRCT domains (*Figure 1B and D*). From the density alone, it is difficult to assign the orientation of the bound DNA at the external site (i.e., which end of the construct is melted). While we cannot assign the exact sequence of the externally bound DNA, we can unambiguously assign the polarity of the strands using the major and minor grooves of DNA. Nevertheless, we establish that there are two DNA binding sites: one internal site that has been recognized since the first RFC structure (*Bowman et al., 2004*), and one external site that we describe here.

The BRCT domain in the N-terminal region of Rfc1 contributes to binding duplex DNA. The BRCT was not visible in any previous structures of RFC (*Bowman et al., 2004*; *Gaubitz et al., 2020*). The BRCT domain docks on the shoulder of Rfc1 where the collar and AAA+ module meet (*Figure 1A and B*) and is connected to the AAA+ module through a long bridging helix. Although BRCT domains are primarily known for acting as protein–protein interaction modules (*Leung and Glover, 2011*), the Rfc1 BRCT domain binds DNA using a surface that has been shown to bind nucleic acids (*Kobayashi et al.,*

*2006*; *Kobayashi et al., 2010*). A series of positively charged and polar residues grip the phosphate backbone of both strands of the DNA duplex (*Figure 1E*). Because there are no interactions with individual bases, the binding mode is consistent with a lack of sequence specificity.

The second DNA binding site has major contributions from the shoulder and collar regions of Rfc1. In this sense, the external DNA binding site is a composite site consisting of interactions from three separate domains. The shoulder of the AAA+ domain exhibits a series of conserved polar and basic residues that grip the duplex, particularly Asn459, Gln474, Arg476, Arg477, Asn481, and Gln508 (*Figure 1F*, *Figure 1—figure supplement 3*). In particular, Gln474 and Arg476 situate into the minor groove of the duplex, potentially setting the register of the bound DNA. Additionally, one of the

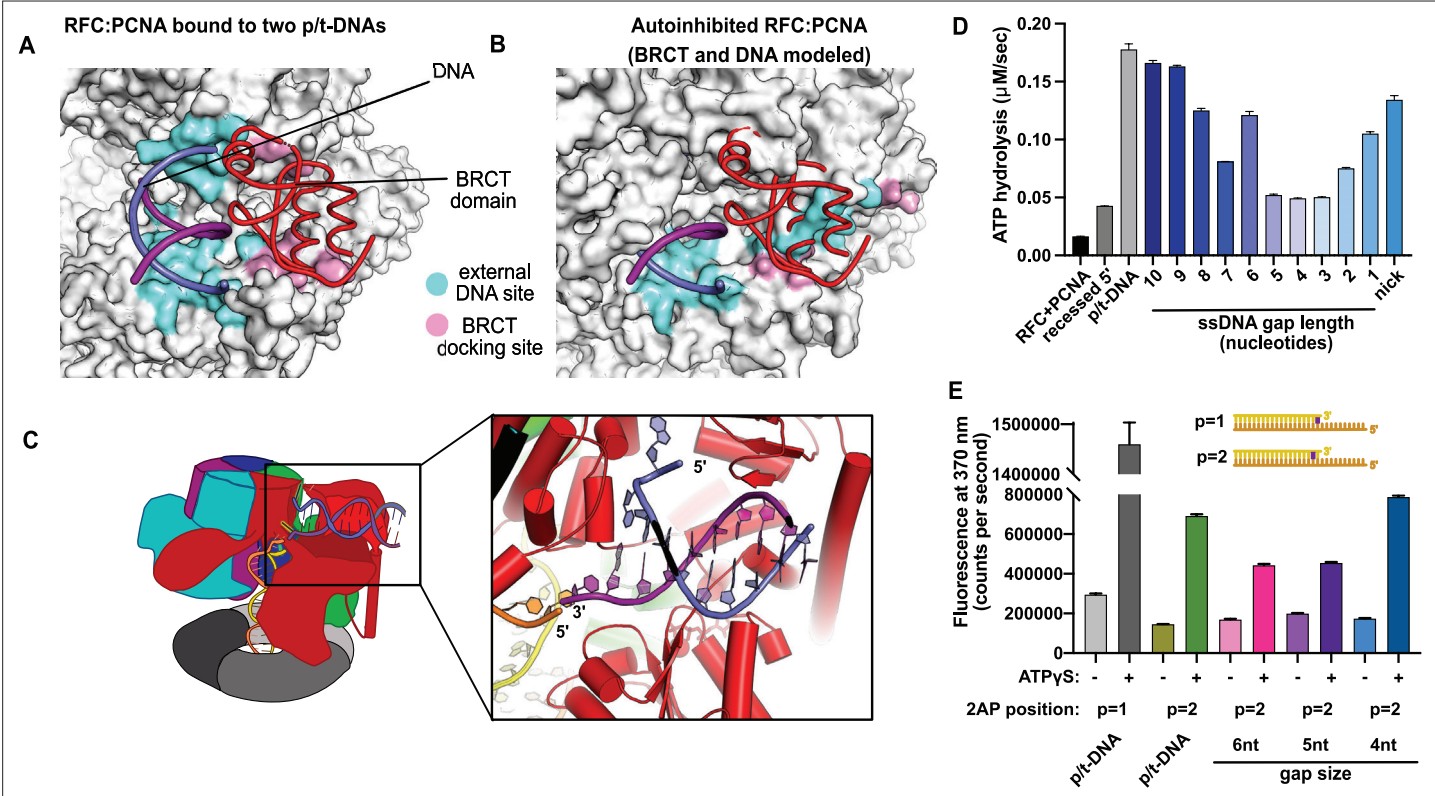

**Figure 2.** Characterization of replication factor C (RFC) utilization of gapped DNA. (**A, B**) The external DNA binding site is incompatible with autoinhibited forms of RFC. (**A**) The structure of replication factor C:proliferating cell nuclear antigen (RFC:PCNA) bound to two p/t-DNAs, highlighting the external DNA binding site. The residues that interact with DNA and the BRCT domain are shown in light blue and pink, respectively. (**B**) The external DNA binding site is disabled in the autoinhibited form of RFC. The BRCT domain and DNA duplex were modeled in the same position relative to the AAA+ fold as shown in panel (**A**). Identical residues of RFC are highlighted in light blue and pink, showing that the binding site is disrupted. Moreover, the collar and A' regions would sterically clash with the DNA and the BRCT domain. (**C**) Because the 5' end of the template strand of the internal DNA is positioned near the 3' end of a strand in the external site, we hypothesized that the two DNA sites could be connected with a short ssDNA gap of approximately 6 nucleotides. (**D**) Steady-state ATPase rates of RFC:PCNA in the presence of p/t-DNA, recessed 5' end DNA, or gapped/nicked DNA. Decreasing gap size results in decreasing ATPase rates until 4 nucleotides of ssDNA, where ATPase rates increase. The trend is smooth and continuous, except for 6-nucleotide single-stranded DNA (ssDNA), which is the size predicted to ideally span the distance between the internal and external sites. (**E**) 2-Aminopurine (2AP) fluorescence measuring base-flipping in gapped DNA constructs. 2AP at the $P = 2$ position informs whether base-flipping occurs at $P = 1$ and/or $P = 2$ position on the primer strand. 0.5 µM RFC and 2 µM PCNA were incubated with 2AP-containing DNA and fluorescence at 370 nm was recorded. We observe ATP-dependent fluorescence changes, particularly in the 4-nucleotide gapped DNA. Error bars in (D&E) reflect the standard deviation from three replicates.

The online version of this article includes the following source data and figure supplement(s) for figure 2:

**Source data 1.** ATPase data.

**Source data 2.** 2-Aminopurine fluorescence.

**Figure supplement 1.** The external DNA binding site is exposed in the open replication factor C:proliferating cell nuclear antigen (RFC:PCNA) structure.

**Figure supplement 2.** Binding of 5' recessed ends to replication factor C (RFC).

strands is placed in between two helix dipoles that have their positive (N-terminal) ends pointed at the backbone of DNA.

This external DNA binding site is only accessible once RFC undergoes the crab-claw opening motion. In RFC's autoinhibited conformation, this region is blocked by the A' domain and the collar domain of the E subunit (Rfc5), which would sterically block six of the nine base pairs of duplex DNA that we observe (*Figure 2A and B*). Moreover, the site where the BRCT domain docks onto the shoulder of Rfc1 is blocked by the collar domains of Rfc1 and the Rfc5. These sites only become accessible after the fold-switching event in Rfc1 that underlies the crab-claw motion for RFC opening (*Gaubitz et al., 2021*; *Figure 2—figure supplement 1*). Therefore, the second DNA binding site and docking of the BRCT domain are coupled to the binding and opening of PCNA.

While the BRCT and AAA+ shoulder regions grip the DNA backbone, the collar domain has several residues that are dedicated to melting and stabilizing the end of DNA (*Figure 1D*). Upon melting, one strand is directed towards a channel between the collar and BRCT domain, while the other strand is directed towards the A-gate. Several residues stabilize the melted bases: Phe552, His556, and Ile664 stack against the duplex DNA and replace the melted bases, and His659 stacks against the melted base directed towards the BRCT domain. Thus, Rfc1 has two separation pins, one in the central chamber and a newly discovered one on the external DNA binding site. This external separation pin is remarkably similar to that used by Rad24-RLC when it binds DNA with a recessed 5' end (*Castaneda et al., 2021*; *Zheng et al., 2021*) (see 'Discussion' for more details).

To test the hypothesis that this external site can bind and melt 5' recessed ends, we used 2-aminopurine (2AP) fluorescence studies to report on base-flipping. 2AP is an adenine analog whose fluorescence is quenched when it is base-paired (*Frey et al., 1995*; *Jean and Hall, 2001*). We previously showed that 2AP fluorescence detects base-flipping at the internal separation pin (*Gaubitz et al., 2021*), and here we employ the same assay to probe the external separation pin. To specifically probe melting at the external separation pin, we used 5' recessed DNA that does not bind the central chamber (*Ellison and Stillman, 2003*), and thus would not be flipped by the internal separation pin. We observe a strong enhancement of 2AP fluorescence when this DNA is mixed with RFC, PCNA, and ATPγS (*Figure 2—figure supplement 2*). This result indicates that the DNA binds and the terminal base of the 5' recessed end is flipped. This fluorescence enhancement is dependent on ATP analog. Because the external DNA binding site is only formed upon ATP-dependent crab-claw opening, this suggests that the melting occurs at this site. Therefore, we confirm that this melting occurs in solution.

The second DNA binding site is intriguingly close to the A-gate, which serves as the exit channel for the template overhang of the internally bound p/t-DNA. The template strand of the internally bound DNA is positioned in similar fashion as we observed in our other structures of yeast RFC bound to p/t-DNA (*Gaubitz et al., 2021*), and the last visible nucleotide of overhang is proximal to the flipped base of the second DNA that is positioned in the A-gate. Moreover, these two strands are in the same polarity (3' to 5'). This led us to hypothesize that the second DNA binding site represents a mechanism for RFC to bind to DNA with single-stranded gaps (*Figure 2C*).

## RFC binding to gapped DNA

We hypothesized that the second DNA binding site promotes RFC binding of duplex DNA with single-stranded gaps. To test this hypothesis, we measured RFC's ATPase rate in the presence of PCNA and various potential DNA substrates. ATPase activity is a reasonable proxy for clamp loading because ATP hydrolysis is triggered in the final stages of the loading reaction (*Anderson et al., 2009*; *Liu et al., 2017*; *Marzahn et al., 2015*; *Sakato et al., 2012*; *Trakselis et al., 2003*).

We observed stimulation of ATPase activity with DNA constructs containing ssDNA gaps ranging from 10 to 1 nucleotide in length, clearly indicating binding of gapped DNA into the central chamber (*Figure 2D*). However, the stimulation across all the gapped substrates is lower than observed for p/t--DNA and is not uniform across all gap sizes. ATPase activity decreases with smaller gap size until it reaches a minimum activity at a 4-nucleotide gap, then increases again at very small gaps and nicked DNA. The ATPase versus gap size profile is remarkably smooth with one glaring exception: the activity from 6-nucleotide gapped DNA is approximately twice as high as one would expect based on the trend. We noted that a 6-nucleotide gap is predicted to be the ideal length to bridge between the internal and external DNA binding sites (*Figure 2C*).

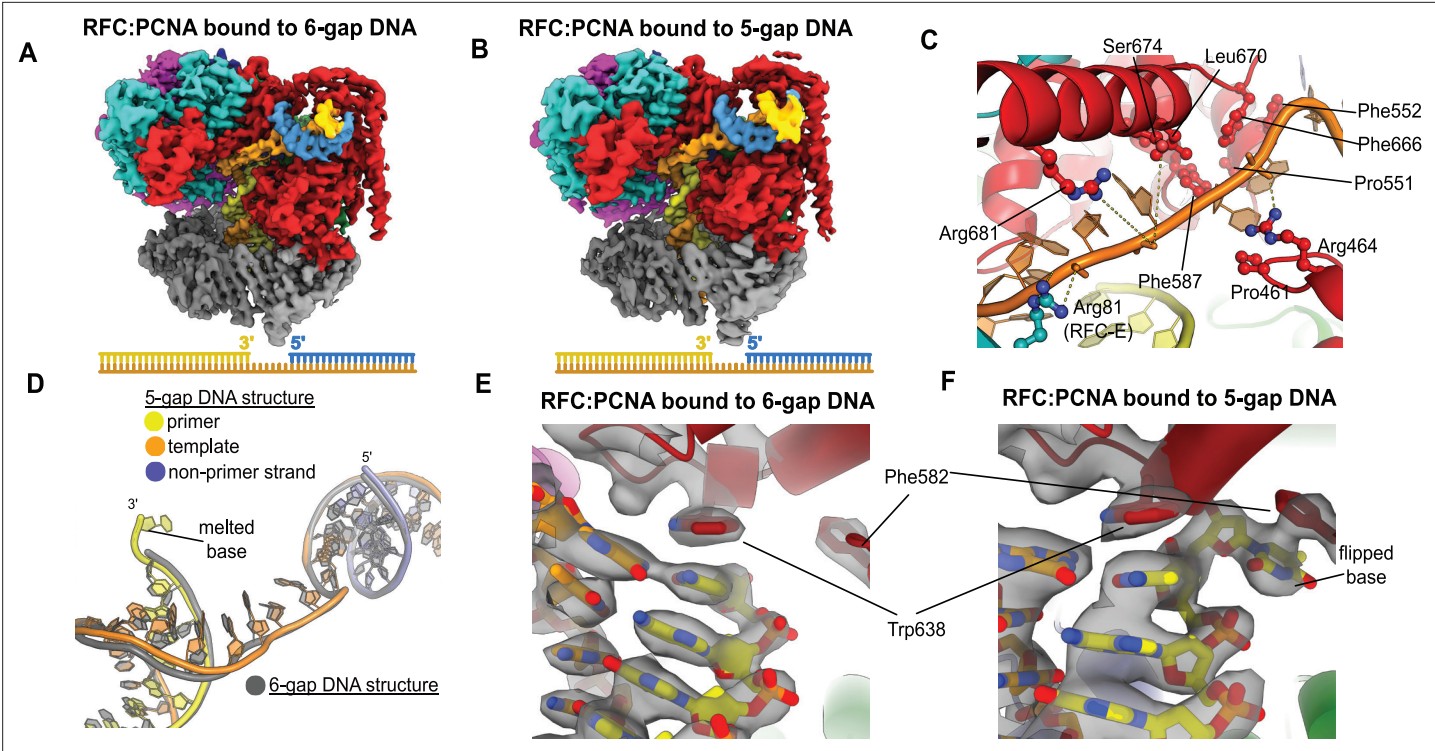

**Figure 3.** Structures of replication factor C:proliferating cell nuclear antigen (RFC:PCNA) bound to gapped DNA. (**A, B**) Cryo-EM reconstruction of the complex of RFC:PCNA bound to DNA with a single-stranded DNA (ssDNA) gap of 6 or 5 nucleotides (6-gap or 5-gap structures). (**C**) The single-strand gap is specifically bound by residues comprising AAA+, collar, and A' domains of the A subunit, as well as contacts from the E subunit. (**D**) DNA conformations in the 5-gap (yellow, orange, and slate) and the 6-gap (gray) structures. The conformations of the DNA are nearly identical except that the 5-gap DNA has melted a single base pair at the internal separation pin so that the ssDNA linker remains 6 nucleotides in length. (**E**) The 6-gap DNA binds with no melting at the internal separation pin. (**F**) The internal separation pin melts a single base pair at the internal separation pin.

The online version of this article includes the following figure supplement(s) for figure 3:

**Figure supplement 1.** Cryo-EM processing of replication factor C:proliferating cell nuclear antigen (RFC:PCNA) in the presence of double-stranded DNA (dsDNA) with a 6-nucleotide gap.

**Figure supplement 2.** Cryo-EM processing of replication factor C:proliferating cell nuclear antigen (RFC:PCNA) in the presence of double-stranded DNA (dsDNA) with a 5-nucleotide gap.

**Figure supplement 3.** RFC binds 5- or 6-gapped DNA without melting at the external separation pin.

We wondered about the relationship between gap size and DNA melting, and whether smaller gap sizes require more melting than larger gap sizes. Therefore, we probed melting activity with gapped DNA labeled with 2AP on the primer strand. We used 2AP adjacent to the 3' end of the primer strand (i.e., the '$P = 2$' position) to probe gapped DNA because we hypothesized that melting of multiple base pairs might be necessary to generate enough ssDNA to span the distance between the internal and external DNA sites. Indeed, we observe significant ATP-dependent fluorescence changes in substrates with ssDNA gaps of 4, 5, and 6 nucleotides (*Figure 2E*). Intriguingly, the fluorescence change is greatest for the 4-nucleotide gapped DNA, suggesting that the construct with the shortest ssDNA segment required the most base-flipping.

To elucidate how RFC binds to gapped DNA, we determined the structures of RFC:PCNA bound to DNA constructs with 6- and 5-nucleotide ssDNA gaps. We chose these gap sizes because we predicted that the 6-nucleotide gap will bind without the need to melt much duplex, while the 5-nucleotide gap should be too short to bind without melting DNA or altering the clamp loader structure to accommodate the shorter linker. We determined the structures of RFC:PCNA bound to 5- and 6-nucleotide gapped DNAs to 3.0- and 3.3 Å overall resolution (*Figure 3A and B*, *Figure 3—figure supplements 1 and 2*).

We find that both structures exhibit the p/t duplex bound in the internal site and the non-p/t duplex bound at the external DNA binding site. In the 5-gap and 6-gap structures, there is clear density for a

single-stranded region of the template strand that connects the two duplex regions (*Figure 3A and B*). The ssDNA linker interacts with a series of mostly conserved residues from the AAA+ (P461, R464), collar (P551, F552, F587), and A′ domains (F666, L670, S674, R681) of the Rfc1, with some additional contributions from the E subunit (S79, R81, N104) (*Figure 3C*, *Figure 3—figure supplement 3*). The ssDNA region traverses the same path that the template overhang does in our structures bound to p/t-DNA (*Figure 3—figure supplement 3A*). These results confirm our hypothesis that the external DNA binding site facilitates binding to DNA with small ssDNA gaps.

The ssDNA region in both the 5-gap and 6-gap structures is 6 nucleotides long, indicating that the 5-gap DNA must have melted at least one base. Indeed, for 5-gap DNA we see clear density for base-flipping of the 3′ end of the primer strand in the internal DNA binding site (*Figure 3D and F*), similar to what we previously observed for p/t-DNA (*Gaubitz et al., 2021*). On the other hand, for 6-gap DNA, we observe scant density for flipping of the 3′ end (*Figure 3D&E*). Neither the 5-gap nor the 6-gap structures show any sign of base-flipping at the external separation pin (*Figure 3—figure supplement 3B and C*).

## RFC binding to nicked DNA

Because ssDNA gaps of 5 or 6 nucleotides in length can be readily accommodated between the two different DNA binding sites without melting at the external separation pin, we asked if RFC can bind nicked DNA. We examined nicked DNA because it is the smallest gap possible. We tested binding of RFC:PCNA to fully duplex DNA containing a single nick using 2AP fluorescence. We placed the 2AP probe at the 3′ end of the primer strand ('$P = 1$' position) to monitor DNA binding and base-flipping in the internal DNA binding site. We observe a large increase in 2AP fluorescence in the presence of RFC:PCNA, greater than the fluorescence increase we observe with standard p/t-DNA (*Figure 4A*). This fluorescence increase is dependent on the presence of ATP analog, indicating that it requires binding to RFC. Therefore, we observe that the RFC:PCNA:ATPγS complex can stably bind to nicked DNA and melt the 3′ end of the primer strand. This effect is not dependent on the presence of a 5′ phosphate in the nonprimer strand, indicating that this moiety is not necessary for the base-flipping at the 3′ end of the primer strand, despite its importance in binding to the BRCT domain (*Allen et al., 1998*; *Kobayashi et al., 2006*).

To reveal how RFC binds to nicked DNA, we determined the cryo-EM structure of RFC:PCNA:ATPγS bound to nicked DNA (with a 5′ phosphate on the non-primer strand) to 3.7 Å overall resolution (*Figure 4B*, *Figure 4—figure supplement 1*, *Table 1*). We again observe duplex DNA bound in both the internal and external sites. Unlike the 5-gap and 6-gap structures that exhibit a 6-nucleotide ssDNA span that links the two duplexes, we observe an ssDNA stretch of 5 nucleotides in the nicked DNA structure (*Figure 4C*). The ssDNA is more taut in the nicked DNA structure, reflecting the mechanical stress that the DNA has undergone to fit within both binding sites.

We observe DNA melting at both separation pins to create the ssDNA gap. We observe clear density for three melted nucleotides at the internal separation pin, with the nucleotides extruded through a cryptic channel that only appears upon RFC opening (*Figure 4D*; *Gaubitz et al., 2021*). We also observe one melted nucleotide at the external separation pin (*Figure 4E*). Therefore, to account for the 5 nucleotides of ssDNA stretching between the two sites, there must be one additional melted base pair that we cannot directly observe. We assigned the nucleotide sequence of the DNA based on size of the density for the bases (i.e., purines vs. pyrimidines), and this assignment matches the melting of 2 nucleotides at the external site. Despite the ambiguity about exactly how much DNA each site can melt, we unambiguously show that the external DNA binding site can also unwind DNA to allow RFC to bind nicked DNA.

The melted region of the nonprimer strand is in a channel formed between the BRCT domain, the AAA+ module, and the collar. This channel is lined with electropositive residues (R174, K190, K209, R511, H556, K557, R600, and H659), most of these residues presumably interact favorably with the backbone of the displaced strand (*Figure 4F*). Like in our 2 p/t-DNA reconstruction, His659's role is for stacking with the base of a melted nucleotide. This residue is conserved as aromatic and most of the other residues are highly conserved as basic in eukaryotes, indicating that both this stacking and electropositive channel are important for function (*Figure 1—figure supplement 3*). The melted nucleotide of the nonprimer strand contacts the BRCT domain (*Figure 4E*). However, the 5′ nucleotide of the nonprimer strand was not resolved well enough to allow for modeling, indicating that the 5′

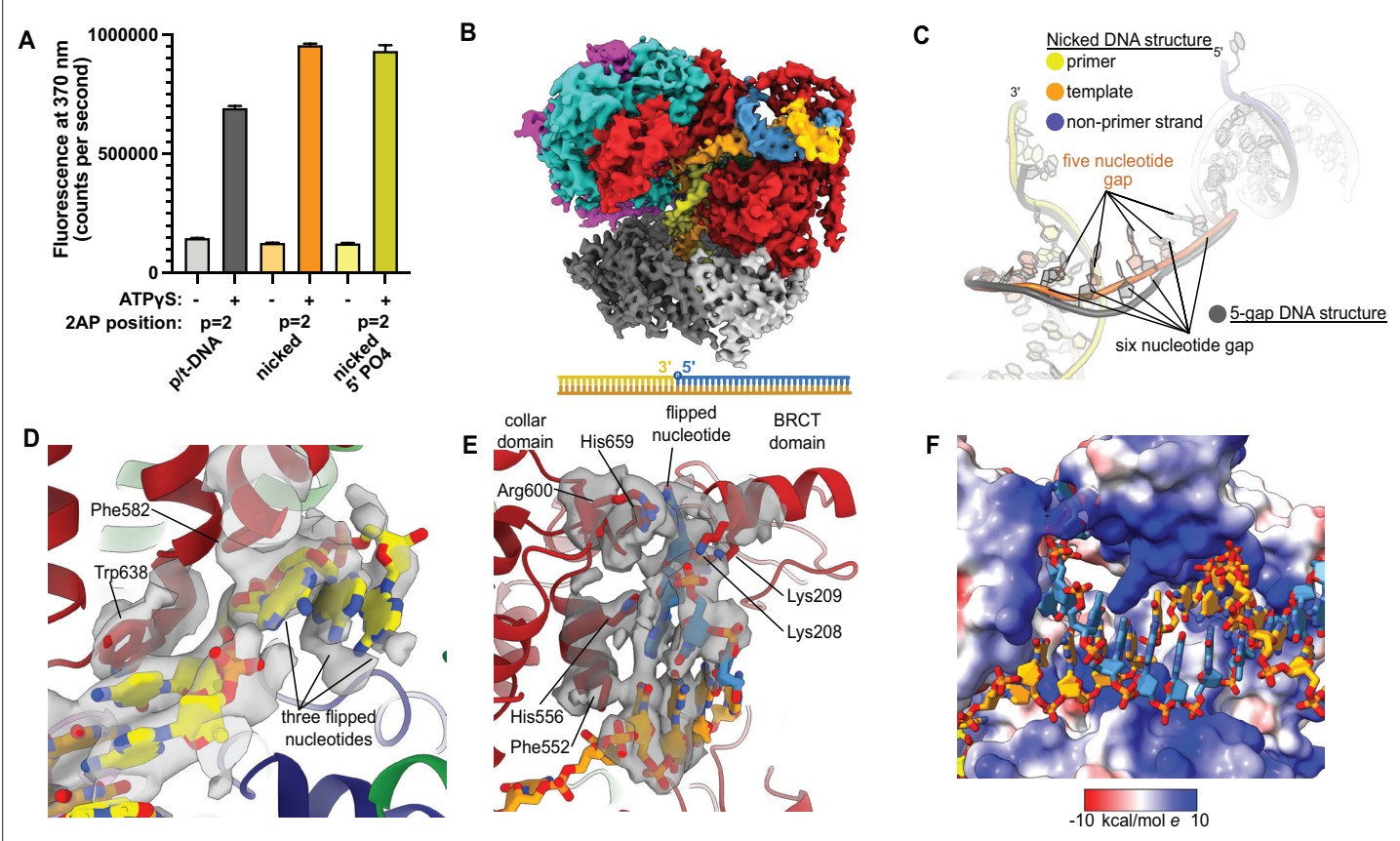

**Figure 4.** Replication factor C:proliferating cell nuclear antigen (RFC:PCNA) binding to nicked DNA. (**A**) RFC melts the primer 3′ end in nicked DNA. 2-Aminopurine (2AP) fluorescence at the primer = 2 position (probe adjacent to the flipped base) showing that the primer strand exhibits even more melting than observed in p/t-DNA. The presence of a 5′ phosphate on the nonprimer strand does not appear to affect base-flipping. Error bars reflect the standard deviation from three replicates. (**B**) Cryo-EM reconstruction of the complex of RFC:PCNA bound to nicked DNA with a 5′ phosphate on the nonprimer strand. (**C**) The nicked DNA contains a single-stranded DNA (ssDNA) gap of five nucleotides linking the two duplex binding sites. This is shorter than the structure with 5-gap DNA (gray), which has a 6-nucleotide-long ssDNA linker. (**D**) At least three base pairs have been melted at the internal separation pin. (**E**) At least one base pair is melted at the external separation pin. (**F**) The flipped nucleotide of the nonprimer strand is in a highly electropositive environment to stabilize the backbone.

The online version of this article includes the following source data and figure supplement(s) for figure 4:

**Source data 1.** 2-Aminopurine fluorescence.

**Figure supplement 1.** Cryo-EM processing of replication factor C:proliferating cell nuclear antigen (RFC:PCNA) in the presence of nicked double-stranded DNA (dsDNA).

**Figure supplement 2.** Effect of 5′ phosphate of the nonprimer strand on replication factor C's (RFC's) ATPase and DNA-melting activity.

**Figure supplement 2—source data 1.** 2-Aminopurine fluorescence.

**Figure supplement 2—source data 2.** ATPase data.

phosphate does not play a major role in the interaction between the DNA bound at the external site and the BRCT domain in the RFC-docked state.

Because the BRCT domain is known to prefer DNA with a 5′ phosphate (*Allen et al., 1998*; *Fotedar et al., 1996*; *Kobayashi et al., 2006*), we tested whether the addition of a 5′ phosphate on the nonprimer strand affects RFC's base-flipping and ATPase activities for nicked or different-sized gapped DNA (*Figure 4—figure supplement 2*). We placed the 2AP probe at the 5′ end of the nonprimer strand ('np = 1' position) to monitor base-flipping in the external DNA binding site. The increase in 2AP fluorescence due to RFC binding is similar when comparing DNA with or without a 5′ phosphate. This result indicates that DNA melting at the external site is unaffected by a 5′ phosphate.

We also compared the effect of a 5′ phosphate on RFC's steady-state ATPase activity using nicked DNA or DNA constructs containing 1–10 nucleotide ssDNA gaps. Overall, the presence of the 5′

**Table 1.** Cryo-EM data collection, processing, and model statistics.

| Dataset | RFC:PCNA with p/t DNA | RFC:PCNA with dsDNA with a 6-nucleotide gap | RFC:PCNA with dsDNA with a 5-nucleotide gap | RFC:PCNA with nicked DNA |
|---|---|---|---|---|
| Magnification | 81,000 | 45,000 | 105,000 | 105,000 |
| Voltage (keV) | 300 | 200 | 300 | 300 |
| Cumulative exposure (e–/Å$^2$) | 40 | 45 | 48 | 49 |
| Detector | K3 | K3 | K3 | K3 |
| Pixel size (Å) | 1.06 | 0.87 | 0.83 | 0.83 |
| Defocus range (μm) | –1.2 to –2.3 | –1–2.2 | –1–2.2 | –1–2 |
| Micrographs used (no.) | 4499 | 4040 | 5118 | 4690 |
| Initial particle images (no.) | 1,331,440 | 797,499 | 1,098,517 | 874,202 |
| Symmetry | C1 | | | |
| Class name | RFC:PCNA bound to two p/t DNA molecules | RFC:PCNA bound to dsDNA with a 6-nucleotide gap | RFC:PCNA bound to dsDNA with a 5-nucleotide gap | RFC:PCNA bound to nicked DNA |
| Final refined particles (no.) | 43,129 | 130,421 | 271,745 | 119,631 |
| Map resolution (Å, FSC 0.143) | 3.4 | 3.3 | 3.0 | 3.7 |
| Model-Map CC_mask | - | 0.83 | 0.82 | 0.8 |
| Bond lengths (Å), angles (°) | - | 0.003, 0.619 | 0.003, 0.596 | 0.003, 0.662 |
| Ramachandran outliers, allowed, favored | - | 0.0, 2.05, 97.95 | 0.0, 1.53, 98.47 | 0.0, 1.79, 98.21 |
| Poor rotamers (%), MolProbity score, Clashscore (all atoms) | - | 0.04, 1.61, 12.23 | 0.04, 1.55, 10.67 | 0.33, 1.60, 12.10 |
| EMDB ID | EMD-26280 | EMD-26298 | EMD-26302 | EMD-26297 |
| PDB ID | - | 7U1A | 7U1P | 7U19 |

RFC, replication factor C; PCNA, proliferating cell nuclear antigen; dsDNA, double-stranded DNA; FSC, Fourier shell correlation.

phosphate on the DNA substrate results in a modest increase to RFC's ATPase activity. The extent of increase in RFC's ATPase activity is not uniform across DNA gap sizes; the increase is more pronounced with larger gaps and less apparent with smaller gaps or nicked DNA. Exactly which steps of the clamp loading process are affected by the addition of the 5′ phosphate is still unclear from our steady-state ATPase activity measurement. Future pre-steady state experiments will investigate this.

We observe no ATP hydrolysis in any of the active sites of the nicked DNA structure, establishing that ATP hydrolysis is not necessary for melting at either the internal or external separation pins. Moreover, the conformation of RFC bound to nicked DNA resembles those bound to 6-gapped and 5-gapped DNA in all other major respects. Because ATP hydrolysis is typically coupled to a conformational change in AAA+ ATPases, this further supports that the melting does not require ATPase activity.

## Deletion of the BRCT domain disrupts response to DNA damage

We hypothesized that the external DNA binding site is used for loading PCNA at gapped or nicked DNA structures in the cell, which typically arise as lp-BER intermediates. These intermediates are structures with either a nick or a small (2–12 nucleotide) gap of ssDNA (*Sattler et al., 2003*), and they require RFC to load PCNA for efficient repair (*Matsumoto, 2001*; *Matsumoto et al., 1999*; *Pascucci et al., 1999*; *Sattler et al., 2003*; *Woodrick et al., 2017*). To interrogate how the BRCT domain affects cellular function, we used an RFC complementation system, where the only copy of RFC1 is supplied on a plasmid. We used a series of mutagen challenges to examine if lp-BER is likely affected

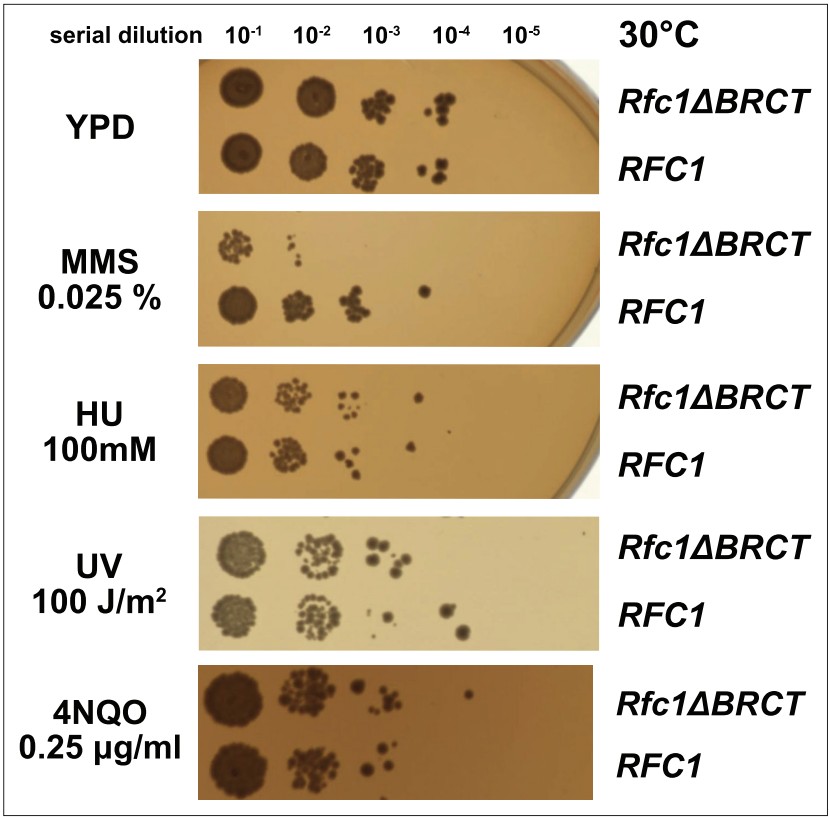

**Figure 5.** Deletion of the BRCT domain of Rfc1 results in a DNA damage repair defect. Yeast carrying the sole copy of the RFC1 gene on a plasmid were subjected to various treatments that stress DNA metabolism. Serial 10-fold dilutions of yeast cultures with a starting OD of 0.2 were spotted onto YPD plates with or without additives and grown for 3 days. Rfc1-ΔBRCT yeast exhibit a growth defect with the DNA alkylating agent methyl methanesulfonate (MMS), but not with hydroxyurea (HU), ultraviolet radiation (UV), or 4-nitroquinoline 1-oxide (4NQO). Various other conditions are shown in *Figure 5—figure supplement 1*.

The online version of this article includes the following figure supplement(s) for figure 5:

**Figure supplement 1.** Deletion of the BRCT domain of Rfc1 results in a DNA damage repair defect.

and if other DNA metabolic pathways are also affected. We used methyl methanesulfonate (MMS; an alkylating reagent that causes damage most commonly repaired by base excision repair), hydroxyurea (HU; which inhibits DNA synthesis by lowering supplies of deoxyribonucleotides), and ultraviolet radiation and 4-nitroquinoline 1-oxide (UV, 4NQO; which cause DNA damage that is usually repaired through nucleotide excision repair) (*Sinha and Häder, 2002*; *Yu and Lee, 2017*; *Matsumoto, 2001*; *Ikenaga et al., 1975*; *Jones et al., 1989*). We found that the RFC1-ΔBRCT construct exhibits a growth defect when in combination with application of the mutagen MMS, where the RFC1-ΔBRCT strain grows approximately 100-fold less well. However, the RFC1-ΔBRCT strain is as sensitive as the wild type to the application of HU, 4NQO, or UV (*Figure 5*). Because melting of DNA is highly dependent on temperature (*Yin et al., 2014*), we repeated the MMS, HU, or UV treatment at various temperatures from 18 to 37°C. The growth defect is seen across a range of MMS concentrations and across a range of growth temperatures, but is not seen with HU and UV at any temperature (*Figure 5—figure supplement 1*). Therefore, we see a growth defect in yeast lacking the Rfc1 BRCT domain that is specific to the damage induced by MMS.

## Discussion

Here, we reveal a second DNA binding site on the external surface of RFC. This site expands RFC's repertoire of targets by facilitating binding to DNA that is either nicked or contains small ssDNA gaps. The external site is composite, consisting of three different domains of Rfc1. This composite nature

allows more flexibility in binding to substrates. Furthermore, the binding site is only found in the open conformation of RFC and is sterically blocked in the autoinhibited conformation, thereby coupling DNA binding to RFC conformation. In these ways, the second DNA binding site has ramifications for the mechanism of substrate recognition, as well as the evolution of alternative clamp loader activities.

## External DNA binding site is similar to that of Rad24-RLC

The external DNA binding site closely resembles the primary DNA binding site in Rad24-RLC. The 'alternative' clamp loader Rad24-RLC differs from RFC by the A-subunit Rfc1, which is swapped out for the Rad24 protein (*Majka and Burgers, 2004*). This seemingly small difference radically changes the loading specificity as Rad24-RLC loads a different sliding clamp (the 9-1-1 complex) onto DNA of opposite polarity (i.e., with a recessed 5′ end) (*Ellison and Stillman, 2003*; *Majka and Burgers, 2004*). Recent structures of Rad24-RLC bound to 9-1-1 and 5′-recessed DNA have revealed the surprising finding that the duplex DNA region is bound on the external surface of the A-subunit Rad24, with the 3′ ssDNA overhang snaking through the A-gate into the central chamber (*Figure 6A*; *Zheng et al., 2021*; *Castaneda et al., 2021*).

Our structures presented here lead to an obvious mechanism for evolution of Rad24-RLC's activity. The classical clamp loader RFC is the likely progenitor of Rad24-RLC. Because Rfc1 already had the ability to bind DNA with a recessed 5′ end, the evolution of Rad24's ability to specifically recognize and load clamps at this DNA architecture is trivial. The DNA binding site of Rad24-RLC also has a region that resembles the external separation pin of Rfc1, although Rad24-RLC has not been observed to melt DNA. It remains to be seen if melting is conserved because such an activity could facilitate loading of 9-1-1 onto the recessed 5′ ends at short ssDNA gaps.

Because the other two known alternative clamp loaders (Ctf18-RLC and Elg1-RLC) also differ from RFC by only the A subunit, our findings highlight that similar evolutionary paths may have led towards their DNA substrate specificities. Future studies will reveal if this is true, and if so, what mechanisms are shared amongst the various clamp loaders.

## RFC has two DNA separation pins

We find that RFC has two separation pins: one in the interior that can melt 3 nucleotides from the primer 3′ end, and an external site that can melt at least one base of DNA. Both separation pins in Rfc1 are conserved across eukaryotes (*Figure 1—figure supplement 3*), indicating that these DNA melting activities are broadly used. Melting at the external site allows for nicked DNA (and perhaps small ssDNA gaps) to bind by creating and/or extending a ssDNA gap that can fit through the A-gate. The external separation pin relies on a channel to funnel the extruded strand. This channel consists of residues from the BRCT domain (K190, K209), the AAA+ module (R511), and the collar domain (H556, K557, R600, and H659) of Rfc1. This composite nature of the external DNA binding site likely makes it more pliable than the internal site, which may allow for novel means of regulation.

Neither separation pin appears to require ATP hydrolysis in order to melt DNA. We previously established that the internal site melts a single base pair without hydrolyzing ATP (*Gaubitz et al., 2021*). While formally possible that melting more than 1 nucleotide requires ATP hydrolysis, our data would argue that this is likely not the case. First, we observe no ATP hydrolysis in the structure of RFC bound to nicked DNA, where five bases are melted. Second, ATP hydrolysis resets the clamp loader structure (*Hingorani and O'Donnell, 1998*; *Turner et al., 1999*), and the open ATP-bound conformation is the only one capable of forming the external DNA binding site. While the idea of an ATPase melting DNA without hydrolyzing ATP might seem unusual, there are numerous examples in the literature of other proteins with this ability. For example, non-NTPase proteins such as CRISPR nucleases and ssDNA binding protein can melt far longer stretches of DNA (*Jore et al., 2011*; *Roy et al., 2009*; *Szczelkun et al., 2014*). Moreover, ATP-dependent helicases can melt substantial amounts of DNA in the absence of ATP hydrolysis (*Lohman and Fazio, 2018*; *Reynolds et al., 2015*). In all of these cases, the energy for melting DNA comes from binding energy. Thus, we propose that RFC also uses binding energy to melt DNA at both separation pins.

## Role for the BRCT domain

Our structures provide the first visualizations of the RFC BRCT domain in the context of an intact clamp loader. We find that the BRCT domain plays an important role in binding DNA at the external

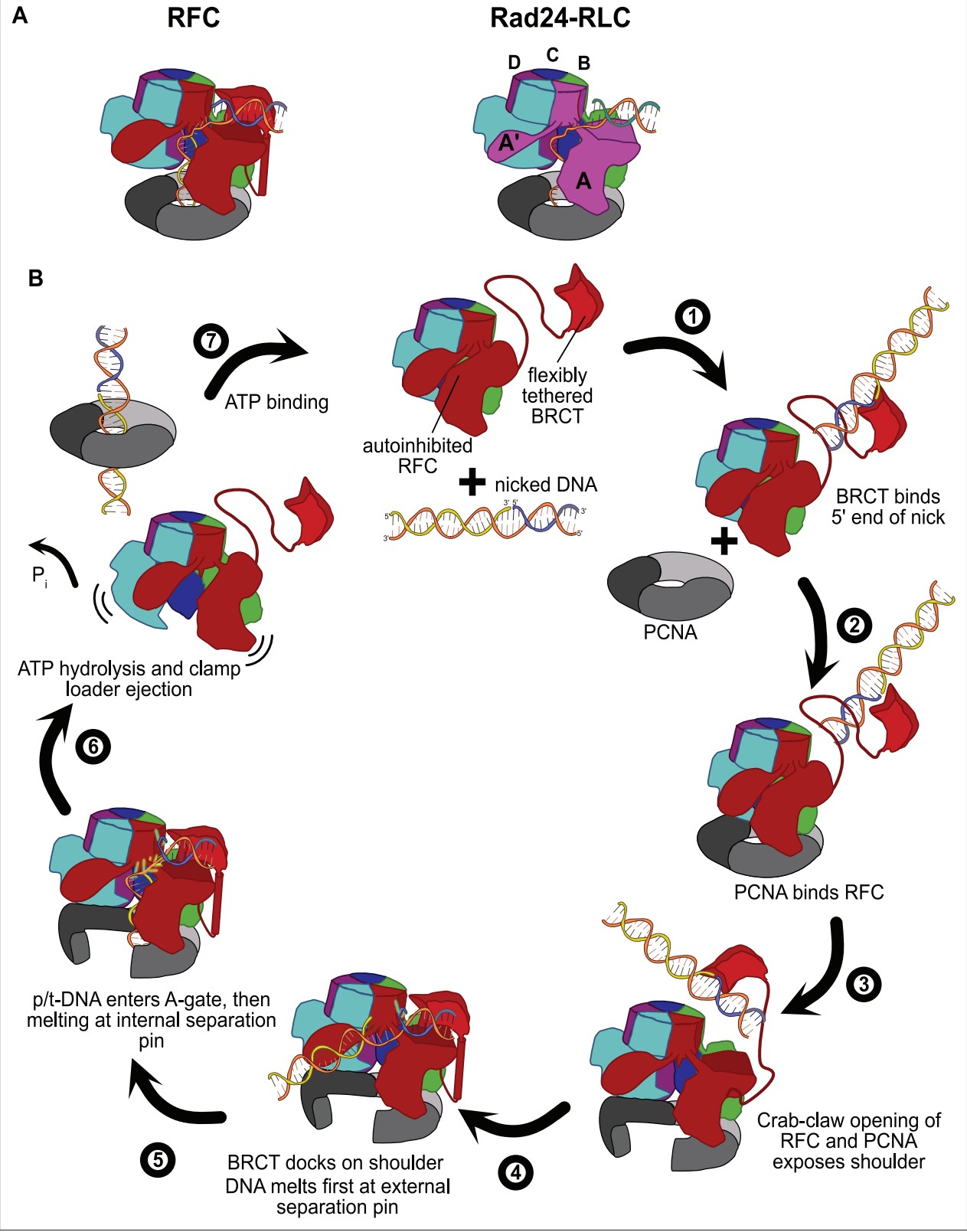

**Figure 6.** Insights into clamp loader evolution and mechanism from the discovery of the external DNA binding site. (**A**) The external binding site of replication factor C (RFC) is similar to that recently reported for Rad24-RLC (*Zheng et al., 2021*; *Castaneda et al., 2021*), suggesting that a straightforward mechanism for the evolution of Rad24-RLC's ability to load clamps at 5'-recessed DNA. (**B**) A speculative model for proliferating cell nuclear antigen (PCNA) loading by RFC onto nicked DNA. Melted bases are shown in glowing green. The loosely tethered BRCT domain can capture

*Figure 6 continued on next page*

*Figure 6 continued*

a DNA segment and keep it in close proximity. Once RFC binds and opens PCNA, the external DNA binding site is formed. DNA melting at the external and internal separation pins allows for nicked DNA to fully engage RFC, thereby activating ATP hydrolysis, clamp closure, and RFC release. See 'Discussion' for a more detailed description.

The online version of this article includes the following figure supplement(s) for figure 6:

**Figure supplement 1.** DNA binding mode of free BRCT is incompatible with the conformation of open replication factor C (RFC).

**Figure supplement 2.** Arginine switch residue is flipped into the active conformation.

DNA binding site. The BRCT domain is docked on top of the AAA+ module of the Rfc1, and is tethered through a long helix (*Figure 1A and B*). The density for the BRCT domain is not as strong as for the other domains of RFC, suggesting that the BRCT domain may sample different orientations. This could provide the external binding site greater pliability in DNA docking, which may provide novel means of regulation. Although it is tempting to claim that this means that the BRCT binds DNA weakly, the isolated BRCT domain binds DNA very tightly, with a $K_d$ of ~10 nM (*Kobayashi et al., 2006*). Therefore, the BRCT domain could act as a flexible tether to initially bind DNA with high affinity before docking onto the core of the clamp loader.

The BRCT domain binds DNA in a different orientation when docked onto RFC than in isolation. Generally, the BRCT domain prefers DNA with a 5′ phosphate end, particularly if there is a 3′ ssDNA overhang (*Allen et al., 1998*; *Fotedar et al., 1996*; *Kobayashi et al., 2006*). A structural model of the isolated BRCT domain bound to DNA was determined using a combination of NMR restraints and modeling in HADDOCK (*Kobayashi et al., 2010*). The 5′ phosphate end binds to the isolated BRCT domain in a deep groove. This phosphate-binding groove is mostly obscured when the BRCT domain is docked onto the shoulder of RFC, and the BRCT domain primarily uses a different face for interacting with DNA (*Figure 1E and F*). Therefore, the DNA binding mode in isolation is incompatible with the conformation of RFC that we observe in our structures as the DNA would clash with the collar domain (*Figure 6—figure supplement 1*). Moreover, the DNA binding mode of the isolated BRCT domain is likely to be only used when the BRCT domain is loosely tethered to the core clamp loader complex. This could provide the BRCT domain the ability to capture DNA at early stages of the loading reaction, possibly before PCNA has bound (*Figure 6B*). To avoid clashing with the collar domain, the BRCT domain must change its grip on DNA upon docking to the AAA+ module. This may play a role in DNA melting and subsequent binding in the central chamber. In support of this hypothesis, the isolated BRCT domain inhibits RFC's loading of PCNA at gapped DNA sites (*Fotedar et al., 1996*).

We find that a 5′ phosphate does not affect RFC's base-flipping activity with 5′ recessed, gapped, or nicked DNA (*Figure 4A*, *Figure 4—figure supplement 2A*). Therefore, the interaction of the BRCT domain with 5′ phosphate is not important for DNA melting at either RFC's internal or external DNA binding sites. Because both assays stall RFC with ATPγS, our 2AP experiments report on a similar intermediate as observed in our cryo-EM structures: a state with PCNA and DNA fully engaged to RFC, but prior to ATP hydrolysis. Therefore, our fluorescence experiments validate that our structures report on physiologically relevant conformations, regardless of whether the DNA has a 5′ phosphate (nicked DNA structure) or does not (5- or 6-gap structures).

In contrast, we observe that the 5′ phosphate has a modest effect on RFC's ATPase activity in the presence of gapped or nicked DNA. Our steady-state ATPase measurements report on multiple states throughout the clamp loading cycle. Previous work has shown that the rate-limiting step in the clamp loading reaction is the release of RFC from DNA (*Sakato et al., 2012*). Taken through this lens, it would appear that our results indicate that the 5′-phosphate accelerates RFC release, while certain sized gaps might decelerate RFC release. However, the study from Sakato et al. focused on clamp loading onto p/t-DNA (*Sakato et al., 2012*), and it remains unknown what the rate-limiting steps are for gapped or nicked DNA. It is quite likely that other steps could be rate-limiting for gapped or nicked DNA, in which DNA recognition and/or melting of DNA is especially challenging. We propose that 5′ phosphate recognition by the BRCT domain is important for rapidly binding DNA substrates with gaps or nicks. Furthermore, we hypothesize that melting of DNA becomes a rate-limiting step for DNA with a nick or small gaps. These hypotheses will be tested by future pre-steady-state experiments.

The BRCT domain is also likely to function as a protein–protein interaction hub, as do most BRCT domains (*Leung and Glover, 2011*). There is evidence that at least two proteins, DNA ligase I and the protease FAM111A, bind to RFC using the BRCT domain (*Hoffmann et al., 2020*; *Levin et al., 2004*). Thus, the coupling of BRCT docking and clamp opening provides a potential mechanism for controlling interactions with RFC partner proteins.

We observe that deletion of the BRCT domain causes a growth defect in yeast grown in the presence of the DNA-alkylating agent MMS (*Figure 5*). Previous work showed that deletion of the entire N-terminal 273 amino acids resulted in sensitivity to MMS (*Gomes et al., 2000*; *McAlear et al., 1996*). This deleted region encompasses the BRCT domain, but also includes several other regions as well. We extend that previous analysis to show that this defect can be specifically localized to the BRCT domain. Furthermore, we observe no growth defect when Rfc1-ΔBRCT yeast are treated with UV or HU.

Based on our nicked and gapped DNA structures along with this phenotype, we propose that the loss of the BRCT domain perturbs lp-BER by disrupting PCNA loading at nicks or small gaps. The damage induced by MMS is primarily repaired by BER (*Lindahl and Wood, 1999*; *Wyatt and Pittman, 2006*), whereas the damage caused by HU and UV is primarily repaired by other pathways (*Petermann et al., 2010*; *Sinha and Häder, 2002*; *Yu and Lee, 2017*). During S phase, the cell primarily relies on the lp-BER subpathway, which requires RFC to load PCNA at nicks and small ssDNA gaps (*Matsumoto, 2001*; *Matsumoto et al., 1999*; *Pascucci et al., 1999*; *Sattler et al., 2003*; *Woodrick et al., 2017*). Because the BRCT domain assists in binding to gapped or nicked DNA, we speculate that the loss of the BRCT domain results in less effective lp-BER. Future experiments will test this hypothesis.

## A speculative model for binding of nicks and small ssDNA gaps

We have synthesized the available biochemical and structural data to derive a mechanistic model for how nicked or gapped DNA structures bind to the clamp loader (*Figure 6B*). The flexibly tethered BRCT domain increases the collision radius of RFC, thereby increasing the likelihood that RFC contacts DNA (*Figure 6B*, step 1). Furthermore, the BRCT domain's intrinsic specificity for the 5′ end of DNA gives it preference in orienting gapped or nicked DNA correctly (*Allen et al., 1998*; *Fotedar et al., 1996*; *Kobayashi et al., 2006*). Binding DNA at the BRCT domain anchors DNA close to the RFC core machinery, thereby increasing the local concentration dramatically. Because the external binding site on the shoulder of Rfc1 is sterically blocked in the autoinhibited conformations, the AAA+ module must bind the sliding clamp prior to the BRCT domain docking onto the shoulder region of RFC (step 2). When the AAA+ module binds PCNA, the complex undergoes the crab-claw opening that both opens the sliding clamp for DNA entry as well as exposes the second DNA binding site on the shoulder of Rfc1 (*Gaubitz et al., 2021*) (step 3). In this way, the clamp loader maintains ordered clamp binding, opening, and loading despite binding to DNA prior to the sliding clamp. After RFC opens, the DNA-bound BRCT domain can now dock onto the shoulder region. The steric clash between the BRCT-bound DNA and the external separation pin necessitates an alteration in the manner in which the BRCT engages DNA (step 4). Thus, docking of DNA-bound BRCT to the shoulder may facilitate unwinding at the external separation pin. Once DNA is fully engaged to the newly formed external DNA binding site, RFC becomes locked into the open conformation and the DNA is co-localized near the A-gate. In this sense, the duplex DNA bound at the external site can act as an allosteric regulator of RFC. Once sufficient DNA is melted at the external site to create a long enough ssDNA linker, duplex DNA then traverses the A-gate into the central chamber. As DNA enters the central chamber, DNA reorients and the 3′ end of the primer melts at the internal separation pin of Rfc1 (step 5). Duplex DNA binding in the chamber activates ATPase activity (step 6), possibly through the flipping of the 'arginine switch.' This conserved arginine holds the catalytic glutamate in an inactive conformation; once DNA binds, the arginine grips DNA and releases the glutamate into an active conformation (*Kelch et al., 2011*). In support of this hypothesis, we observe that the arginine switch residue is flipped into the active conformation in most of our structures (*Figure 6—figure supplement 2*). The DNA also rigidifies the AAA+ spiral, which is likely a key factor in ATPase activity (*Subramanian et al., 2021*). Finally, ATP hydrolysis and $P_i$ release triggers clamp closure and ejection of the clamp loader, which can then recycle to load another clamp (step 7). Our model makes numerous predictions for the clamp loading reaction that will need to be tested in future studies.

# Materials and methods

**Key resources table**

| Reagent type (species) or resource | Designation | Source or reference | Identifiers | Additional information |
|---|---|---|---|---|
| Strain, strain background (*Escherichia coli*) | BL21(DE3) | Novagen | 69450 | Chemically competent cells |
| Recombinant DNA reagent | pET(11a)-RFC[2+3+4] (plasmid) | *Finkelstein et al., 2003* | | Expression plasmid |
| Recombinant DNA reagent | pLANT-2/RIL[1+5] (plasmid) | *Finkelstein et al., 2003* | | Expression plasmid |
| Recombinant DNA reagent | pRS413-RFC1 (plasmid) | *Gaubitz et al., 2021* | | Plasmid for yeast expression of Rfc1 from endogenous promoter |
| Recombinant DNA reagent | pRS413-RFC1RFC1-ΔBRCT (plasmid) | This study | | Plasmid for yeast expression of Rfc1 from endogenous promoter |
| Strain, strain background (*Saccharomyces cerevisiae*) | BY4743 his3Δ1/his3Δ1 leu2Δ0/leu2Δ0 LYS2/lys2Δ0 met15Δ0/MET15 ura3Δ0/ura3Δ0 Δrfc1::KanMX4/RFC1 (YOR217W) | Dharmacon | YSC1055 (22473) | Yeast Heterozygous Collection |
| Software, algorithm | RELION | doi: 10.7554/eLife.42166 | Relion 3.1 | |
| Software, algorithm | cisTEM | doi: 10.7554/eLife.35383 | cisTEM-1.0.0-beta | https://cistem.org/software |
| Software, algorithm | Ctffind | doi: 10.1016j.jsb.2015.08.008 | Ctffind 4.1 | |
| Software, algorithm | UCSF Chimera | UCSF, doi: 10.1002/jcc.20084 | | http://plato.cgl.ucsf.edu/chimera/ |
| Software, algorithm | ChimeraX | UCSF, doi: 10.1002/pro.3943 | ChimeraX-1.2 | https://www.cgl.ucsf.edu/chimerax/ |
| Software, algorithm | COOT | doi:10.1107/S0907444910007493 | Coot-0.9.4 | http://www2.mrc-lmb.cam.ac.uk/personal/pemsley/coot/ |
| Software, algorithm | Phenix | doi:10.1107/S0907444909052925 | Phenix-1.20.1–4487 | https://phenix-online.org |
| Software, algorithm | PyMOL | PyMOL Molecular Graphics System, Schrodinger LLC | | https://www.pymol.org/ |
| Software, algorithm | GraphPad Prism | GraphPad | GraphPad Prism 9.2.1 | http://www.graphpad.com/ |
| Peptide, recombinant protein | Pyruvate kinase | Calzyme | 107A0250 | Enzyme used in ATPase assay |
| Peptide, recombinant protein | Lactate dehydrogenase | Worthington Biochemical Cooperation | LS002755 | Enzyme used in ATPase assay |
| Other | Phosphoenol-pyruvic acid monopotassium salt | Alfa Aesar | B20358 | Reagent used in ATPase assay |
| Chemical compound, drug | Methyl methanesulfonate (MMS) | Sigma-Aldrich | 66-27-3 | https://www.sigmaaldrich.com/US/en/product/aldrich/129925 |
| Chemical compound, drug | Hydroxyurea (HU) | Sigma-Aldrich | 127-07-1 | https://www.sigmaaldrich.com/US/en/product/sigma/h8627 |
| Chemical compound, drug | 4-Nitroquinoline (4NQO) | Fisher Scientific | AC203790010 | https://www.fishersci.com/shop/products/4-nitroquinoline-n-oxide-98-thermo-scientific-3/AC203790010 |

## Protein expression and purification

RFC was purified as described previously (*Gaubitz et al., 2021*). Briefly, pET(11a)–RFC[2+3+4] and pLANT-2/RIL–RFC[1+5] were transformed into BL21(DE3) *Escherichia coli* cells (Millipore). For protein expression, transformants were grown in 4 L of prewarmed terrific broth medium with 50 µg/mL kanamycin and 100 µg/mL ampicillin at 37°C and induced with IPTG at an optical density of 0.8. Expression was continued at 18°C for 16–18 hr. Cells were pelleted and resuspended in 300 mL lysis

buffer (30 mM HEPES–NaOH pH 7.5, 250 mM NaCl, 0.25 mM EDTA, 5% glycerol, 2 mM DTT, 2 µg/mL aprotinin, 0.2 µg/mL pepstatin, 2 µg/mL leupeptin, 1 mM PMSF). RFC was purified by chromatography over 10 mL SP-Sepharose (80 mL gradient of 300–600 mM NaCl) and 10 mL Q-Sepharose (GE Healthcare, 40 mL gradient of 150–500 mM NaCl). Fractions of RFC were pooled and dialyzed overnight into 30 mM HEPES–NaOH pH 7.5, 250 mM NaCl, 5% glycerol, and 2 mM DTT.

PCNA was purified as described previously (*Gaubitz et al., 2021*). Briefly, BL21 (DE3) *E. coli* cells were transformed with a 6xHis-PPX-PCNA-expressing pET-28 vector (PPX = Precission protease). For protein expression, 1 L of induced cells was grown overnight at 18°C in terrific broth medium with 50 µg/mL kanamycin. Cells were pelleted and resuspended 30 mM HEPES pH 7.6, 20 mM imidazole, 500 mM NaCl, 10% glycerol, and 5 mM β-mercaptoethanol. Upon cell lysis, centrifugation, and lysate filtration, the lysate was applied to a 5 mL HisTrap FF column (GE Healthcare). The column was washed with a 1 M NaCl buffer and subsequently with a 50 mM NaCl buffer. PCNA was eluted with 500 mM imidazole. The eluted protein was cleaved with Precission protease. The cleaved protein was applied to a 5 mL HiTrap Q HP column (GE Healthcare), from which it was eluted with 2 M NaCl in a 100 mL gradient. RFC-containing fractions were dialyzed against 30 mM Tris pH 7.5, 100 mM NaCl, and 2 mM DTT. Purified RFC was concentrated, aliquoted, and frozen in liquid nitrogen for storage at –80°C.

## Crosslinking
RFC and PCNA were mixed in a 1:1 molar ratio and buffer exchanged into 1 mM TCEP, 200 mM NaCl, 25 mM HEPES–NaOH, pH 7.5, and 4 mM $MgCl_2$. The protein complex was diluted to 3 µM, and 1 mM ATPγS was added to the protein complex and incubated for 2 min. Subsequently, 7 µM 5-gapped, 6-gapped, or nicked DNA was added and incubated for another 1 min. The sequences of the DNA oligonucleotides are listed in *Table 2*. The crosslinking reaction was started with 1 mM BS3, incubated for 15 min at room temperature, and neutralized with Tris–HCl.

## Electron microscopy
### Cryo-EM sample preparation
Quantifoil R 0.6/1 grids were washed with ethyl acetate, and glow discharged with Pelco easiGlow for 60 s at 25 mA (negative polarity). 2.8 µL of the sample with p/t DNA and 3.5 µL of the samples with gapped or nicked DNA were applied to grids at 10°C and 95% humidity in a Vitrobot Mark IV (FEI). Samples were blotted with a force of 5 for 5 s after a 2 s wait and plunged into liquid ethane.

### Cryo-EM data collection
RFC:PCNA with p/t-DNA was imaged on a Titan Krios operated at 300 kV and equipped with a K3 detector at ×81,000 magnification and a pixel size of 0.53 Å in super-resolution mode. 4499 micrographs were collected with a target defocus of –1.2 to –2.3 and a total exposure of ~40 e-/Å$^2$ per micrograph averaging 30 frames.

RFC:PCNA:6-gapped DNA was imaged on a Talos Arctica at 200 kV, at ×45,000 magnification and a pixel size of 0.435 (bin = 0.5), using a K3 detector in super-resolution mode. The data was collected with a target defocus range of –1 to –2.2 µm, and a total exposure of ~45 e-/Å$^2$. 4039 micrographs were recorded using the 'multi-shot' method, applying image shift and beam tilt to collect one shot per hole and nine holes per stage move in ~17.5 hr with SerialEM (*Mastronarde, 2003*). The dose rate is 22.56 electrons/unbinned pixel/s.

Both RFC:PCNA:5-gapped DNA and RFC:PCNA:nicked DNA were imaged on a Titan Krios operated at 300 kV and equipped with a Gatan energy filter at a slit width of 20 eV at ×105,000 magnification and a pixel size of 0.415 Å (bin = 0.5), using a K3 detector in super-resolution mode.

The RFC:PCNA:5-gapped DNA data was collected with a target defocus range of –1 to –2.2 µm, and a total exposure of 47.7 e-/Å$^2$. Multi-shot of 1 × 15 (single shot × 15 holes) was used to record 5121 micrographs in ~25 hr with a dose rate of 24.70 electrons/unbinned pixel/s.

The RFC:PCNA:Nicked DNA data was collected with a target defocus range of –1 to –2 µm, and a total exposure of 48.7 e-/Å$^2$. Multi-shot of 1 × 9 (single shot × 9 holes) was used to record 5651 micrographs in ~19 hr with a dose rate of 24.70 electrons/unbinned pixel/s.

**Table 2.** DNA sequences.

| Template | Sequence | Primer | Sequence | Nonprimer | Sequence | Name in assay |
|---|---|---|---|---|---|---|
| Template30, T30 | TTTTTTTTTATGTA CTCGTAGTGTCTGC | Primer20-2AP-0 | GCAGACACTACG AGTACAT/32AmPu/ | | | p/t-DNA $P = 1$ |
| Template30-T-1 | TTTTTTTTTTGTA CTCGTAGTGTCTGC-3' | Primer20-2AP-1 | GCAGACACTACG AGTACA/i2AmPr/A | | | p/t-DNA $P = 2$ |
| Template50-ap_gapped2 | TTGTGGGTAGAT AAATACAGACCTAA GTCCTTTGTA CTCGTAGTGTCTGC | Primer20-2AP-1 | GCAGACACTACG AGTACA/i2AmPr/A | 3'PrimerB24_gapped | AGGTCTGTATTT ATCTACCCACAA | 6 nt gap $P = 2$ |
| Same as above | | Same as above | | 3'PrimerB25_gapped | TAGGTCTGTATTT ATCTACCCACAA | 5 nt gap $P = 2$ |
| Same as above | | Same as above | | 3'PrimerB26_gapped | TTAGGTCTGTATTT ATCTACCCACAA | 4 nt gap $P = 2$ |
| Same as above | | Same as above | | 3'PrimerB30_gapped | GGACTTAGGTCTGTA TTTATCTACCCACAA | Nicked $P = 2$ |
| Same as above | | Same as above | | 3'PrimerB30_gapped_P | /5Phos/GGACTTA GGTCTGTA TTTATCTACCCACAA | Nicked 5' $PO_4$ $P = 2$ |
| Template30-3'-T | TATGTACTCGTAGTG TCTGTTTTTTTTTT | | | Primer20-2AP-20 | /52AmPr/CAGACA CTACGAGTACATA | Recessed 5' $P = 1$ |
| Template50-ap_nick_np1 | TTGTGGGTAGATA AATACAGACC TAAGTCTTATGTAC TCGTAGTGTCTGC | Primer20-2AP-0 | GCAGACACTACG AGTACAT/32AmPu/ | | | p/t-DNA $P = 1$ (design 2, used for 2AP + $PO_4$) |
| | | | | 3'PrimerB30_np1 | AGACTTAGGTCTGTA TTTATCTACCCACAA | Recessed 5' np = 1 |
| Template50-ap_nick_np1 | TTGTGGGTAGATAA ATACAGACC TAAGTCTTATGTACT CGTAGTGTCTGC | Primer20-2AP-0 | GCAGACACTACG AGTACAT/32AmPu/ | 3'PrimerB30_np1 | AGACTTAGGTCTGTA TTTATCTACCCACAA | Nicked np = 1 |
| Template50-ap_6gap_np1 | TTGTGGGTAGATA AATACAGACCT AAGTCTTTTTTAT GTACTCGTAGTGTCTGC | Same as above | | Same as above | | 6 nt gap np = 1 |

*Table 2 continued on next page*

*Table 2 continued*

| Template | Sequence | Primer | Sequence | Nonprimer | Sequence | Name in assay |
|---|---|---|---|---|---|---|
| Template50-ap_5gap_np1 | TTGTGGGTAGATA AATACAGACCTAAG TCTTTTTTATGTA CTCGTAGTGTCTGC | Same as above | | Same as above | | 5 nt gap np = 1 |
| Template50-ap_4gap_np1 | TTGTGGGTAGATA AATACAGACCTAA GTCTTTTTATGTA CTCGTAGTGTCTGC | Same as above | | Same as above | | 4 nt gap np = 1 |
| | | | | 3'PrimerB30_2AP_np1_P | /5Phos//i2AmPr/GAC TTAGGTCTGTA TTTATCTACCCACAA | Nonprimer np = 1 + PO4 (2AP) |
| Template50_gapped | TTGTGTGGGTAGATAA ATACAGACCTAA GTCCTTGAATGCC GCGTGCGTCCC | 5'Primer 20_gapped | GGGACGCACGC GGCATTCAA | 3'PrimerB20_gapped | CTGTATTTATCTACCCACAA | p/t-DNA (used in ATPase assay) |
| Same as above | | Same as above | | Same as above | | 10 gap |
| Same as above | | 5'Primer 21_gapped | GGGACGCACGC GGCATTCAAG | Same as above | | 9 gap |
| Same as above | | 5'Primer22_gapped | GGGACGCACGCG GCATTCAAGG | Same as above | | 8 gap |
| Same as above | | 5'Primer23_gapped | GGGACGCACGCG GCATTCAAGGA | Same as above | | 7 gap |
| Same as above | | 5'Primer24_gapped | GGGACGCACGC GGCATTCAAGGAC | Same as above | | 6 gap (ATPase, cryo-EM) |
| Same as above | | 5'Primer25_gapped | GGGACGCACGC GGCATTCAAGGACT | Same as above | | 5 gap (ATPase, cryo-EM) |
| Same as above | | 5'Primer26_gapped | GGGACGCACGCGGC ATTCAAGGACTT | Same as above | | 4 gap |
| Same as above | | 5'Primer27_gapped | GGGACGCACGCGGC ATTCAAGGACTTA | Same as above | | 3 gap |
| Same as above | | 5'Primer28_gapped | GGGACGCACGCGGC ATTCAAGGACTTAG | Same as above | | 2 gap |

*Table 2 continued on next page*

*Table 2 continued*

| Template | Sequence | Primer | Sequence | Nonprimer | Sequence | Name in assay |
|---|---|---|---|---|---|---|
| Same as above | | 5'Primer29_gapped | GGGACGCCACGCGGC ATTCAAGGACTTAGG | Same as above | | 1 gap |
| Same as above | | 5'Primer30_gapped | GGGACGCGCACGCGGC ATTCAAGGACTTAGGT | Same as above | | Nicked |
| | | | | 3'PrimerB20_gapped_P | /5Phos/CTGTATTTA TCTACCCACAA | Nonprimer with 5' phosphate (ATPase) |
| Primer20-3'-T-10ext | GCAGACACTACGAG TACATTTTTTTTTTT | Template20-5'-A | AATGTACTCGT AGTGTCTGC | | | Recessed 5' |
| Template50-ap_gapped2 | TTGTGGGTAGATAA ATACAGACCTAAG TCCTTTGTACTCG TAGTGTCTGC | Primer20-1 | GCAGACACTA CGAGTACAAA | 3'PrimerB30_gapped_P | /5Phos/GGACTTAGGTCTGT ATTTATCTACCCACAA | Nicked 5' PO$_4$ (cryo-EM) |

## Data processing

Data processing for the RFC:PCNA:2p/tDNA was performed exactly as described previously (*Figure 1—figure supplement 1*; *Gaubitz et al., 2021*). Data processing of the datasets with nicked or gapped DNA was performed with minor modifications as follows (*Figure 3—figure supplements 1 and 2* and *Figure 4—figure supplement 1*). Micrograph frames were aligned in IMOD (*Kremer et al., 1996*) with 2× binning, resulting in a pixel size of 0.83 Å/pixel. Particle picking was performed using cisTEM (*Grant et al., 2018*). Following particle picking, particles were extracted with a box size of 240 pixels (dataset with p/t DNA) or 320 pixels (all other datasets) and subjected to 2D classification. (*Figure 3—figure supplement 1A and B*, *Figure 3—figure supplement 2A and B*, and *Figure 4—figure supplement 1A and B*). Particles from classes with well-defined features were selected for processing in Relion 3.1. Coordinates and micrographs were imported into Relion 3.1 (*Zivanov et al., 2018*), CTF parameters were re-estimated with CtfFind-4.1 (*Rohou and Grigorieff, 2015*), and particles were subjected to several rounds of 3D classification. For 3D classification of the dataset with the 6-gapped DNA, the down-filtered RFC:PCNA:2p/tDNA cryo-EM reconstruction was used as reference, for 3D classification of the datasets with the 5-gapped and nicked DNA, the downfiltered reconstructions of RFC:PCNA:6-gapped DNA and RFC:PCNA:5-gapped DNA, respectively, were used as reference (*Figure 3—figure supplements 1C and 2C*, *Figure 4—figure supplement 1C*). All references were downfiltered to 50 Å. In the dataset of RFC:PCNA bound to 6-gapped DNA, we observe classes of RFC bound to opened or closed PCNA. However, for further refinement we focused on classes that represented the highest number of particles and also had the highest resolution. In these classes, PCNA is always in the closed conformation across the different datasets. The cryo-EM density was postprocessed in Relion for estimating the resolution and autosharpened (for RFC:PCNA:2p/tDNA) or density modified with PHENIX for model building and refinement (*Terwilliger et al., 2020*; *Table 1*). Model information was not used for density modification.

## Model building and refinement

The structure of yeast RFC bound to closed PCNA and p/t DNA (PDB ID: 7TID) was used for initial fitting of the cryo-EM reconstruction of RFC:PCNA bound to 6-gapped DNA. All subunits were split into globular domains and fitted into the cryo-EM density with UCSF Chimera (*Pettersen et al., 2004*). For building the BRCT domain and the adjacent alpha-helix, the AlphaFold model (*Jumper et al., 2021*; *Varadi et al., 2022*) corresponding to these residues was fitted into the cryo-EM density. The model was adjusted in Coot (*Emsley and Cowtan, 2004*), and the DNA was built in manually. The manually adjusted model of RFC:PCNA:6-gapped DNA was used to rigid body fit the cryo-EM density of RFC:PCNA bound to 5-gapped DNA and the density of RFC:PCNA bound to nicked DNA. The fitted models were adjusted and the DNA was built in manually in Coot. The adjusted models were real-space refined in PHENIX1.19 and PHENIX1.20.1 (*Liebschner et al., 2019*). UCSF Chimera and PyMOL were used for figure generation (*Delano, 2002*; *Pettersen et al., 2004*; *Figure 3—figure supplements 1D and 2D*, *Figure 4—figure supplement 1D*, *Table 1*).

The density of RFC:PCNA bound to two p/tDNA was rigid body fitted with the model for RFC:PCNA bound to 5-gapped DNA. The DNA was replaced by p/t DNA and an ideal B-form DNA molecule for the externally bound DNA. The flipped bases in proximity to the external separation pin were added manually with Coot. The density did not allow the unambiguous assignment of the exact sequence of the externally bound DNA. Moreover, the PCNA density in proximity to RFC-E is highly distorted, likely owing to a structural heterogeneity of particles in this class with PCNA being bound to RFC in the open and closed form. Therefore, the model was not further refined.

## ATPase assays

0.12 µM RFC was incubated with a master mix (3 U/mL pyruvate kinase, 3 U/mL lactate dehydrogenase, 1 mM ATP, 670 µM phosphoenol pyruvate, 170 µM NADH, 50 mM Tris [pH 7.5], 0.5 mM TCEP, 5 mM MgCl$_2$, 200 mM potassium glutamate, 40 mM NaCl), 1 µM PCNA, and 1 µM annealed DNA substrates, whose sequences are listed in *Table 2*. ATPase activity was measured at room temperature with the 2014 VICTOR Nivo Multimode Microplate Reader to detect NAD+. Rates were obtained from a linear fit of the slopes using GraphPad Prism. For each data point, three experimental replicates were performed.

## 2AP fluorescence

2AP fluorescent samples were excited at 315 nm (5 mm slit width), and emission was detected at 370 nm (7 mm slit width) with a FluoroMax 4 (Horiba Jobin Yvon Inc). Reactions contained 200 nM annealed DNA, 0.5 µM RFC, and 2 µM PCNA in a solution of 50 mM HEPES–NaOH pH 7.5, 200 mM NaCl, 4 mM MgCl₂, 1 mM TCEP, and were carried out at room temperature. 1 mM ATPγS was added to the reaction, and measurement was taken after 4 min of incubation. For each data point, three experimental replicates were performed.

## Plasmid generation

The complementation plasmid pRS413-RFC1 contains the entire RFC1 sequence, where Rfc1 is expressed under the control of its own promoter. The ΔBRCT RFC variant was introduced with site-directed mutagenesis in pRS413-RFC1, resulting in pRS413-Rfc1ΔBRCT. Residues 154–230 in RFC1 were replaced by a GSGS linker.

## Yeast strain and spot assay

The *Saccharomyces cerevisiae* strain used in this study for transformation with pRS413-RFC1 and pRS413-Rfc1ΔBRCT and subsequent dissection was obtained from the Dharmacon Yeast Heterozygous Collection and verified by PCR. The genotype of BY4743 is his3Δ1/his3Δ1 leu2Δ0/leu2Δ0 LYS2/lys2Δ0 met15Δ0/MET15 ura3Δ0/ura3Δ0 Δrfc1::KanMX4/RFC1 (YOR217W). *S. cerevisiae* culture, transformation, and tetrad dissection were performed as described by *Gomes et al., 2000*.

For the spot assay, yeast was grown on a plate with synthetic medium without histidine at 30°C for 2 days, subsequently inoculated into 3 mL synthetic complete-His media and grown for 3–4 hr to an OD of 0.8. Serial 10-fold dilutions of the culture starting from OD of 0.2 were plated as 4 µL drops onto YPD plates with or without chemical additives (0.015, 0.02, 0.025% MMS, 100, 200 mM HU, 0.05, 0.1, and 0.25 µg/mL 4NQO). For UV treatment, the spotted yeast was irradiated with 30 or 100 J/m² using a UVP UV crosslinker. The plates were imaged after incubating at 18°C for 7 days, or at 30 or 37°C for 3 days. Three replicates were done for all conditions.

## Acknowledgements

The authors thank Drs. C Xu, KK Song, K Lee, and C Ouch for assistance with data collection, and Dr. C Xu for advice on data processing. We thank Drs. T Kubota and N Rhind as well as members of the Kelch, Royer, Munson, Rando, and Schiffer labs for helpful discussions. We thank Dr. Mary Munson for technical support for yeast experiments. This work was funded by NIGMS (R01-GM127776). CG was supported by an Early and Advanced Postdoc Mobility (grant numbers 168972 and 177859) Fellowship of the Swiss National Science Foundation.

## Additional information

### Funding

| Funder | Grant reference number | Author |
| --- | --- | --- |
| National Institute of General Medical Sciences | 1R01GM127776-01A1 | Brian A Kelch |
| Swiss National Science Foundation | 177859 | Christl Gaubitz |

The funders had no role in study design, data collection and interpretation, or the decision to submit the work for publication.

### Author contributions

Xingchen Liu, Joshua Pajak, Conceptualization, Data curation, Formal analysis, Investigation, Validation, Visualization, Writing - original draft, Writing - review and editing; Christl Gaubitz, Conceptualization, Data curation, Formal analysis, Funding acquisition, Investigation, Methodology, Project administration, Supervision, Validation, Visualization, Writing - original draft, Writing - review and

editing; Brian A Kelch, Conceptualization, Formal analysis, Funding acquisition, Project administration, Resources, Software, Supervision, Validation, Visualization, Writing - original draft, Writing - review and editing

**Author ORCIDs**
Xingchen Liu ![ORCID] http://orcid.org/0000-0002-9089-1761
Christl Gaubitz ![ORCID] http://orcid.org/0000-0002-6047-9282
Brian A Kelch ![ORCID] http://orcid.org/0000-0002-1369-6989

**Decision letter and Author response**
Decision letter https://doi.org/10.7554/eLife.77483.sa1
Author response https://doi.org/10.7554/eLife.77483.sa2

---

## Additional files

**Supplementary files**
• Transparent reporting form

**Data availability**
The reported cryo-EM map and atomic coordinates have been deposited in the Electron Microscopy Data Bank (entry numbers EMD-26280 EMD-26298 EMD-26302 EMD-26297) and the Protein Data Bank (ID codes 7U1A, 7U1P, 7U19).

The following datasets were generated:

| Author(s) | Year | Dataset title | Dataset URL | Database and Identifier |
|---|---|---|---|---|
| Gaubitz L, Kelch P | 2022 | RFC bound to PCNA and two primer/template DNA molecules | https://www.ebi.ac.uk/emdb/26280 | EMDataResource, 26280 |
| Gaubitz L, Kelch P | 2022 | RFC:PCNA bound to dsDNA with a ssDNA gap of six nucleotides | https://doi.org/10.2210/pdb7U1A/pdb | Worldwide Protein Data Bank, 10.2210/pdb7U1A/pdb |
| Gaubitz L, Kelch P | 2022 | RFC:PCNA bound to DNA with a ssDNA gap of five nucleotides | https://doi.org/10.2210/pdb7U1P/pdb | Worldwide Protein Data Bank, 10.2210/pdb7U1P/pdb |
| Gaubitz L, Kelch P | 2022 | RFC:PCNA bound to nicked DNA | https://doi.org/10.2210/pdb7U19/pdb | Worldwide Protein Data Bank, 10.2210/pdb7U19/pdb |

---

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
