## [Editor Report]

This study leverages the fortuitous finding that the eukaryotic RFC clamp loader binds to two DNA molecules at distinct sites. The first at the previously described internal binding site and the second, novel, site at a site on RFC1 external to the loader and above (viewed from a certain perspective) the PCNA ring. The authors extend and validate their chance finding using biochemical, genetic, and structural studies. The cryo-EM structures of RFC bound to nicked DNA that emerge from this study write a new chapter in the biochemistry of clamp loading.

---

## [Decision Letter]

**Decision letter after peer review:**

Thank you for submitting your article "A second DNA binding site on RFC facilitates clamp loading at gapped or nicked DNA" for consideration by *eLife*. Your article has been reviewed by 3 peer reviewers, including Bruce Stillman as the Reviewing Editor and Reviewer #1, and the evaluation has been overseen Volker Dötsch as the Senior Editor. The following individuals involved in the review of your submission have agreed to reveal their identity: David Jeruzalmi (Reviewer #2); Peter Burgers (Reviewer #3).

Essential revisions:

A major concern with this study is the choice of DNA substrate. Previous biochemical studies from several labs have shown that binding of 5'-junction DNA to an isolated BRCT domain strongly depends on the presence of the 5'-phosphate. Other single BRCT domains that bind DNA, e.g. from Rev1, also show a strong dependence on the 5'-phosphate. DNA repair intermediates, such as base excision repair products after incision by Apn1/2, carry 5'-phosphates. Very surprisingly, the DNA substrates used in this study lack the physiologically relevant 5'-phosphate. The only experiment in the paper that indirectly addresses the issue is in Figure 4A; it shows that the melting of the 5'-nucleotide occurs independently of the presence of the phosphate. There is no discussion on why the authors chose the unphosphorylated DNA substrate. If the phosphate indeed is an important feature, it would benefit the authors to determine cryoEM studies with the proper DNA.

1. Page 2 in the introduction, left paragraph. The authors cite papers showing that Rad24-RLC binds to ssDNA/dsDNA with a 5'-recessed ssDNA/dsDNA junction and cite Kubota et al. 2015, and two papers from Majka and Burgers. This was first shown with the human Rad24-RFC by Ellison and Stillman in 2003 and this reference could be cited here.

2. The authors show that mutations that alter the amino-terminus of Rfc1 create sensitivity to MMS. They should cite the following paper that has reported genetic evidence that mutations in RFC1 cause sensitivity to MMS. These are not the same mutations, but nevertheless, confirm the new finding:

McAlear, M. A., Tuffo, K. M. and Holm, C. The large subunit of replication factor C (Rfc1p/Cdc44p) is required for DNA replication and DNA repair in *Saccharomyces cerevisiae*. Genetics 142, 65-78 (1996).

3. The RFC subunits have proper names, Rfc1 through Rfc5. Please, indicate how A and E subunits, etc., correspond to those. Authors also tend to switch up and down between nomenclatures which is confusing to the reader.

4. P6L, middle. The ATPase assay (Figure 2D) is described too vaguely, and conclusions of the gap size effect are lacking. This steady-state ATPase assay measures multiple turnovers, which requires iterative cycles of PCNA loading, followed by PCNA sliding (away from the loading site and off the DNA) and RFC dissociation, before reloading. The lower ATPase of the 5-nt gap would be consistent with the more sustained binding of the complex.

5. The DNA damage sensitivity study of the δ-BRCT mutant closely recapitulates that of a Rfc1 NTD deletion study over twenty years ago, which was quoted. With the added insights in DNA repair pathways in general, and with the BRCT DNA binding properties here specifically, a more targeted site-directed mutagenesis approach would be more appropriate, and so would an epistasis analysis. For example, is the BRCT-δ phenotype epistatic to that of an Apn1/2 deletion? In that regard, it is of interest to note that Apn2 has a PCNA binding motif.

*Reviewer #1 (Recommendations for the authors):*

1. Page 2 in the introduction, left paragraph. The authors cite papers showing that Rad24-RLC binds to ssDNA/dsDNA with a 5'-recessed ssDNA/dsDNA junction and cite Kubota et al. 2015, and two papers from Majka and Burgers. This was first shown with the human Rad24-RFC by Ellison and Stillman in 2003 and this reference could be cited here.

2. The authors show that mutations that alter the amino-terminus of Rfc1 create sensitivity to MMS. They should cite the following paper that has reported genetic evidence that mutations in RFC1 cause sensitivity to MMS. These are not the same mutations, but nevertheless, confirm the new finding

McAlear, M. A., Tuffo, K. M. and Holm, C. The large subunit of replication factor C (Rfc1p/Cdc44p) is required for DNA replication and DNA repair in *Saccharomyces cerevisiae*. Genetics 142, 65-78 (1996).

*Reviewer #2 (Recommendations for the authors):*

Could the authors quantitate the outcomes in figure 5? What dilutions were used? (This information is in the methods, but better to not force the reader to look for necessary information).

In supplement Figure 6.1. may I suggest that the authors avoid the use of the names Watson and Crick to designate individual strands of DNA? This nomenclature is entirely honorific but without analytical purpose. Bestowing honor to the accomplishments of scientists through the use of their names is a very laudable goal. However, in such cases, careful efforts should be made to bestow honor in an inclusive manner. In the case of efforts to understand the structure of double-stranded DNA, the complex interplay between Watson, Crick, and Franklin should be recognized and acknowledged.

*Reviewer #3 (Recommendations for the authors):*

1. The RFC subunits have proper names, Rfc1 through Rfc5. Please, indicate how A and E subunits, etc., correspond to those. Authors also tend to switch up and down between nomenclatures which is confusing to the reader.

2. P6L, middle. The ATPase assay (Figure 2D) is described too vaguely, and conclusions of the gap size effect are lacking. This steady-state ATPase assay measures multiple turnovers, which requires iterative cycles of PCNA loading, followed by PCNA sliding (away from the loading site and off the DNA) and RFC dissociation, before reloading. The lower ATPase of the 5-nt gap would be consistent with the more sustained binding of the complex.

3. The DNA damage sensitivity study of the δ-BRCT mutant closely recapitulates that of an Rfc1 NTD deletion study over twenty years ago, which was quoted. With the added insights in DNA repair pathways in general, and with the BRCT DNA binding properties here specifically, a more targeted site-directed mutagenesis approach would be more appropriate, and so would an epistasis analysis. For example, is the BRCT-δ phenotype epistatic to that of an Apn1/2 deletion? In that regard, it is of interest to note that Apn2 has a PCNA binding motif.

---

## [Author Response]

Essential revisions:A major concern with this study is the choice of DNA substrate. Previous biochemical studies from several labs have shown that binding of 5'-junction DNA to an isolated BRCT domain strongly depends on the presence of the 5'-phosphate. Other single BRCT domains that bind DNA, e.g. from Rev1, also show a strong dependence on the 5'-phosphate. DNA repair intermediates, such as base excision repair products after incision by Apn1/2, carry 5'-phosphates. Very surprisingly, the DNA substrates used in this study lack the physiologically relevant 5'-phosphate. The only experiment in the paper that indirectly addresses the issue is in Figure 4A; it shows that the melting of the 5'-nucleotide occurs independently of the presence of the phosphate. There is no discussion on why the authors chose the unphosphorylated DNA substrate. If the phosphate indeed is an important feature, it would benefit the authors to determine cryoEM studies with the proper DNA.

We agree with the reviewers that the 5’-phosphate is physiologically relevant. We thank the reviewers for pointing out that we did not report this clearly in our manuscript. Actually, the nicked DNA substrate we used for our cryo-EM study has a 5’-phosphate on its non-primer strand (Table 2). To clarify this better, we adjusted the text in the Results section and in Figure 4B to indicate that the 5’-phosphate is present on the non-primer strand. All changes in the text of the manuscript are highlighted in blue.

In the density map of RFC:PCNA bound to nicked DNA, the 5’ nucleotide of the non-primer strand was not resolved well, so we did not model it in our final structure. The 5-gap and 6-gap DNA substrates do not have the 5’-phosphate, and their 5’ end is not poised to interact with BRCT domain in the structures with RFC similar to what we observed in the nicked DNA structure.

Nevertheless, we wanted to explore whether the 5’-phosphate is important for DNA melting and PCNA loading. Therefore, we measured 2AP fluorescence using DNA with or without 5’phosphate (Figure 4—figure supplement 2). We placed the 2AP in the non-primer strand to report on base-flipping activity of RFC at the external DNA binding site. The addition of 5’ phosphate did not significantly change the base flipping activity of RFC on 5’-recessed, 5- or 6-nucleotide gapped DNA, and nicked DNA. This base-flipped state that the 2AP experiments report on is presumably the same state as that of our cryoEM structures. With a gap of ideal length (6nt), the DNA is capped on each end and fits the clamp loader without the need for melting in either the internal nor the external DNA binding site. With a 5nt gap, one nucleotide melts and we observe in the structure that this is occurring primarily on the interior site. We do not observe melting at the exterior site, in line with the results of our 2AP experiments (Figure 4—figure supplement 2). Taken together, we conclude that the 5’ phosphate is not important for melting activity.

We further probed the importance of the 5’-phosphate using steady state ATPase measurements. The addition of the 5’ phosphate triggers a modest increase in ATPase activity of RFC for some of the gapped DNA substrates, but not nicked DNA. However, we cannot conclude which step of the clamp loading reaction is affected by the 5’ phosphate from our steady state ATPase measurements. Future pre-steady state experiments will investigate this further, as these experiments are beyond the scope of this study. Taken together, our 2AP and ATPase measurements indicate that the gapped DNA structures that we captured by cryo-EM are not significantly perturbed by the lack of a 5’-phosphate. The new results are shown in Figure 4— figure supplement 2 as well as described in the Results and Discussion sections.

1. Page 2 in the introduction, left paragraph. The authors cite papers showing that Rad24-RLC binds to ssDNA/dsDNA with a 5'-recessed ssDNA/dsDNA junction and cite Kubota et al. 2015, and two papers from Majka and Burgers. This was first shown with the human Rad24-RFC by Ellison and Stillman in 2003 and this reference could be cited here.

We have updated the text to cite this paper in the Introduction.

2. The authors show that mutations that alter the amino-terminus of Rfc1 create sensitivity to MMS. They should cite the following paper that has reported genetic evidence that mutations in RFC1 cause sensitivity to MMS. These are not the same mutations, but nevertheless, confirm the new finding:McAlear, M. A., Tuffo, K. M. and Holm, C. The large subunit of replication factor C (Rfc1p/Cdc44p) is required for DNA replication and DNA repair in *Saccharomyces cerevisiae*. Genetics 142, 65-78 (1996).

We have updated the text to cite this paper in the Discussion.

3. The RFC subunits have proper names, Rfc1 through Rfc5. Please, indicate how A and E subunits, etc., correspond to those. Authors also tend to switch up and down between nomenclatures which is confusing to the reader.

We thank the reviewers for pointing this out. We have updated Figure 1 to indicate how each subunit of RFC corresponds to their gene name.

4. P6L, middle. The ATPase assay (Figure 2D) is described too vaguely, and conclusions of the gap size effect are lacking. This steady-state ATPase assay measures multiple turnovers, which requires iterative cycles of PCNA loading, followed by PCNA sliding (away from the loading site and off the DNA) and RFC dissociation, before reloading. The lower ATPase of the 5-nt gap would be consistent with the more sustained binding of the complex.

We thank the reviewers for pointing this out. We have updated the Results and Discussion sections to provide a more thorough description of the ATPase results and their interpretation. We agree with the reviewers that RFC’s lower ATP hydrolysis rate with 5-nt gap DNA could be explained by a slower RFC release from the DNA. Indeed, release of RFC from primer-template DNA is thought to be the rate-limiting step under steady-state conditions (Sakato et al. 2012). However, all of the prior pre-steady state measurements were performed using p/t-DNA, but not with nicked or gapped DNA. Therefore it is also possible that other steps in the clamp loading mechanism prior to RFC release (initial DNA binding, DNA melting etc.) become rate-limiting when RFC loads at gapped or nicked DNA. Future pre-steady state experiments will investigate this hypothesis in more detail.

5. The DNA damage sensitivity study of the δ-BRCT mutant closely recapitulates that of a Rfc1 NTD deletion study over twenty years ago, which was quoted. With the added insights in DNA repair pathways in general, and with the BRCT DNA binding properties here specifically, a more targeted site-directed mutagenesis approach would be more appropriate, and so would an epistasis analysis. For example, is the BRCT-δ phenotype epistatic to that of an Apn1/2 deletion? In that regard, it is of interest to note that Apn2 has a PCNA binding motif.

We agree that the specific role of the BRCT domain warrants further investigation. We now report that rfc1-∆BRCT yeast cells grow equally well as WT when challenged with the DNA damaging reagent 4NQO (Figure 5 and Figure 5—figure supplement 1). These results support the notion that deletion of the BRCT domain causes yeast to have defective base excision repair, because 4NQO damage is primarily repaired by NER. Therefore, our new data strengthen our hypothesis that the Rfc1 BRCT domain primarily facilitates RFC’s functions in base excision repair.

We fully agree that further mutagenesis and epistasis studies will be important for teasing apart the precise role of the BRCT domain in RFC function; this will be investigated in future studies and is beyond the scope of this manuscript.

Reviewer #1 (Recommendations for the authors):1. Page 2 in the introduction, left paragraph. The authors cite papers showing that Rad24-RLC binds to ssDNA/dsDNA with a 5'-recessed ssDNA/dsDNA junction and cite Kubota et al. 2015, and two papers from Majka and Burgers. This was first shown with the human Rad24-RFC by Ellison and Stillman in 2003 and this reference could be cited here.

As mentioned above, we have cited this paper.

2. The authors show that mutations that alter the amino-terminus of Rfc1 create sensitivity to MMS. They should cite the following paper that has reported genetic evidence that mutations in RFC1 cause sensitivity to MMS. These are not the same mutations, but nevertheless, confirm the new findingMcAlear, M. A., Tuffo, K. M. and Holm, C. The large subunit of replication factor C (Rfc1p/Cdc44p) is required for DNA replication and DNA repair in *Saccharomyces cerevisiae*. Genetics 142, 65-78 (1996).

As mentioned above, we have cited this paper.

Reviewer #2 (Recommendations for the authors):Could the authors quantitate the outcomes in figure 5? What dilutions were used? (This information is in the methods, but better to not force the reader to look for necessary information).

We thank the reviewer for this helpful suggestion to improve this figure. We have updated the text and the figure to describe the results more quantitatively.

In supplement Figure 6.1. may I suggest that the authors avoid the use of the names Watson and Crick to designate individual strands of DNA? This nomenclature is entirely honorific but without analytical purpose. Bestowing honor to the accomplishments of scientists through the use of their names is a very laudable goal. However, in such cases, careful efforts should be made to bestow honor in an inclusive manner. In the case of efforts to understand the structure of double-stranded DNA, the complex interplay between Watson, Crick, and Franklin should be recognized and acknowledged.

We agree and we have changed the names of the DNA strands of the second DNA molecule bound on RFC’s exterior DNA binding site to strand1 and strand2 in Figure 1 and Figure 6—figure supplement 1.

Reviewer #3 (Recommendations for the authors):1. The RFC subunits have proper names, Rfc1 through Rfc5. Please, indicate how A and E subunits, etc., correspond to those. Authors also tend to switch up and down between nomenclatures which is confusing to the reader.

We have updated Figure 1 and the manuscript for clarification and to keep the nomenclature consistent.

2. P6L, middle. The ATPase assay (Figure 2D) is described too vaguely, and conclusions of the gap size effect are lacking. This steady-state ATPase assay measures multiple turnovers, which requires iterative cycles of PCNA loading, followed by PCNA sliding (away from the loading site and off the DNA) and RFC dissociation, before reloading. The lower ATPase of the 5-nt gap would be consistent with the more sustained binding of the complex.

As mentioned above, we have updated the manuscript to include a more thorough discussion of the steady state ATPase results.

3. The DNA damage sensitivity study of the δ-BRCT mutant closely recapitulates that of an Rfc1 NTD deletion study over twenty years ago, which was quoted. With the added insights in DNA repair pathways in general, and with the BRCT DNA binding properties here specifically, a more targeted site-directed mutagenesis approach would be more appropriate, and so would an epistasis analysis. For example, is the BRCT-δ phenotype epistatic to that of an Apn1/2 deletion? In that regard, it is of interest to note that Apn2 has a PCNA binding motif.

As mentioned above, we have challenged δ BRCT yeast with an additional DNA damaging reagent to further test the hypothesis that RFC’s BRCT domain contributes to BER. A systematic study with yeast double mutants is a fantastic suggestion and will be the focus of follow-up work.